# ABCC9-related Intellectual disability Myopathy Syndrome is a K$_{ATP}$ channelopathy with loss-of-function mutations in ABCC9

Marie F. Smeland[1,16]*, Conor McClenaghan[2,16], Helen I. Roessler [3,16], Sanne Savelberg[3], Geir Åsmund Myge Hansen[1], Helene Hjellnes[1], Kjell Arne Arntzen[4,5,6], Kai Ivar Müller [4,5], Andreas Rosenberger Dybesland[6,7], Theresa Harter[2], Monica Sala-Rabanal[2,14], Chris H. Emfinger[2], Yan Huang[2,15], Soma S. Singareddy[2], Jamie Gunn[8], David F. Wozniak[8], Attila Kovacs[9], Maarten Massink[3], Federico Tessadori [3,10], Sarah M. Kamel [10], Jeroen Bakkers [10,11], Maria S. Remedi[12], Marijke Van Ghelue[1,13,17], Colin G. Nichols [2,17] & Gijs van Haaften[3,17]*

Mutations in genes encoding K$_{ATP}$ channel subunits have been reported for pancreatic disorders and Cantú syndrome. Here, we report a syndrome in six patients from two families with a consistent phenotype of mild intellectual disability, similar facies, myopathy, and cerebral white matter hyperintensities, with cardiac systolic dysfunction present in the two oldest patients. Patients are homozygous for a splice-site mutation in ABCC9 (c.1320 + 1 G > A), which encodes the sulfonylurea receptor 2 (SUR2) subunit of K$_{ATP}$ channels. This mutation results in an in-frame deletion of exon 8, which results in non-functional K$_{ATP}$ channels in recombinant assays. SUR2 loss-of-function causes fatigability and cardiac dysfunction in mice, and reduced activity, cardiac dysfunction and ventricular enlargement in zebrafish. We term this channelopathy resulting from loss-of-function of SUR2-containing K$_{ATP}$ channels ABCC9-related Intellectual disability Myopathy Syndrome (AIMS). The phenotype differs from Cantú syndrome, which is caused by gain-of-function ABCC9 mutations, reflecting the opposing consequences of K$_{ATP}$ loss- versus gain-of-function.

[1] Department of Medical Genetics, University Hospital of North Norway, 9019 Tromsø, Norway. [2] Department of Cell Biology and Physiology, and Center for the Investigation of Membrane Excitability Diseases (CIMED), Washington University, St Louis, MO 63110, USA. [3] Department of Genetics, Center for Molecular Medicine, University Medical Center Utrecht, 3584 CX Utrecht, the Netherlands. [4] Department of Neurology, University Hospital of North Norway, 9019 Tromsø, Norway. [5] Department of Clinical Medicine, UiT—The Arctic University of Norway, 9019 Tromsø, Norway. [6] The National Neuromuscular Centre of Norway, University Hospital of North Norway, 9019 Tromsø, Norway. [7] Department of Physiotherapy, University Hospital of North Norway, 9019 Tromsø, Norway. [8] Department of Psychiatry, Washington University School of Medicine, St. Louis, MO 63110, USA. [9] Department of Medicine, Washington University School of Medicine, St. Louis, MO 63110, USA. [10] Hubrecht Institute-KNAW and UMC Utrecht, 3584 CT Utrecht, the Netherlands. [11] Department of Medical Physiology, Division of Heart and Lungs, University Medical Center Utrecht, 3584 CX Utrecht, the Netherlands. [12] Department of Medicine, Division of Endocrinology, Metabolism and Lipid Research, Washington University, St Louis, MO 63110, USA. [13] Department of Medical Genetics, the Arctic University of Norway, 9019 Tromsø, Norway. [14]Present address: Department of Anesthesiology, Washington University, St Louis, MO 63110, USA. [15]Present address: Department of Cardiology, Renmin Hospital of Wuhan University, Wuhan, China. [16]These authors contributed equally: Marie F. Smeland, Conor McClenaghan, Helen I. Roessler. [19]These authors jointly supervised this work: Marijke Van Ghelue, Colin G. Nichols, Gijs van Haaften. *email: marie.smeland@unn.no; G.vanHaaften@umcutrecht.nl

K$_{ATP}$ channels are nucleotide-gated potassium channels formed by the obligate co-assembly of pore-forming Kir6.x subunits and regulatory sulfonylurea receptors (SURx), which couple the membrane potential to metabolic state in multiple cell types[1,2]. In mammals, two Kir6.x isoforms are encoded by the paralogous *KCNJ8* (Kir6.1; [OMIM: 600935]) and *KCNJ11* (Kir6.2; [OMIM: 600937]) genes, which are each co-located with genes encoding two SUR isoforms, *ABCC9* (SUR2; [OMIM: 601439]) and *ABCC8* (SUR1; [OMIM: 600509]), respectively, on chromosomes 12 and 11. Molecular heterogeneity is further increased by alternative splicing of *ABCC9* mRNA, yielding two major splice variants, SUR2A and SUR2B— while multiple other splice variants have also been reported[1,3–8]. Pancreatic and neuronal K$_{ATP}$ channels are predominantly formed by Kir6.2 and SUR1, smooth muscle K$_{ATP}$ channels are comprised of Kir6.1 and SUR2B, and the predominant combination in striated muscle is Kir6.2 and SUR2A[3].

The causative role of gain-of-function (GoF) or loss-of-function (LoF) mutations in the Kir6.2/SUR1-dependent pancreatic K$_{ATP}$ channels in neonatal diabetes and congenital hyperinsulinism, respectively, was established nearly two decades ago[9–12]. Recently, it has been demonstrated that dominant GoF mutations in *KCNJ8* and *ABCC9* underlie Cantú Syndrome (CS [OMIM: 239850])[13–15]. CS is characterized by hypertrichosis, coarse facial features, and multiple cardiovascular abnormalities, including cardiomegaly and tortuous, dilated vasculature[14–16]. Behavioral problems and mild developmental delay have been reported in CS, but intellectual function is typically normal[17].

The human consequences of LoF in Kir6.1 and SUR2 remain uncertain. In a single report, two heterozygous LoF mutations in an exon found only in SUR2A were associated with dilated cardiomyopathy (DCM [MIM: 608569])[18]. A missense mutation in the same exon was reported as predisposing to paroxystic adrenergic atrial fibrillation (AF [MIM: 614050]), but only in one 53-year-old female patient[19]. The pathophysiological consequences of complete SUR2 LoF are unclear.

We report six patients from two non-consanguineous families from Northern Norway who exhibit a shared pathological constellation including similar facies, intellectual disability and developmental delay, anxiety, myopathy with hypotonia, muscle weakness, and fatigability. Cardiac systolic dysfunction is found in the two oldest patients. All have cerebral white matter hyperintensities, and hyperreflexia is found in the oldest four. The families are investigated by comprehensive clinical exome sequencing, a powerful tool for identifying the genetic basis of rare and complex syndromes, both in patients with de novo mutations and in families with suspected recessive inheritance[20,21].

We identify a homozygous *ABCC9* splice site mutation (c.1320 + 1 G > A) in all affected individuals. We show that the mutation causes the in-frame deletion of exon 8, resulting in SUR2 protein lacking 52 amino acids, and loss of plasmalemmal K$_{ATP}$ function. Using CRISPR/Cas9 genome engineering, we introduce frameshift mutations into *ABCC9* that result in premature protein truncation, in both zebrafish and mice. These animals lack functional SUR2 protein and myocyte K$_{ATP}$ channels and recapitulate the myopathy and cardiac dysfunction observed in patients. We conclude that SUR2 LoF results in a recessive syndrome: *ABCC9*-related Intellectual disability Myopathy Syndrome (AIMS).

## Results

**Patient descriptions**. Four siblings from Family 1 (aged 12–21 years) and two siblings from Family 2 (aged 29, 33) (Fig. 1a) are described. The families are not known to be related, but are from

the same area of Northern Norway. Genetic investigations had earlier been performed with normal results in several of the patients, including G-banding, high-resolution Single-Nucleotide Polymorphism (SNP) array to look for genomic deletions and duplications, FMRI CGG repeat analysis, DMPK PCR, sequencing of multiple neuromuscular disease genes, screening for mitochondrial DNA sequence variants/deletions, and screening for inborn errors of metabolism. Clinical photographs of the patients are presented in Fig. 1b, c. MRI images are presented in Fig. 1d. Clinical features are summarized in Tables 1 and 2.

**Family 1**. *Patient 1–1* is the first child of healthy parents from Northern Norway with probable Finnish ancestry. Pregnancy and birth were uneventful. Weight and length were below the 2.5 percentile in childhood. Hearing loss was reported in childhood, but a recent hearing evaluation was normal. Early psychomotor development was described as normal, but at age 2, in-toeing, toe-walking and reduced fine motor skills, generalized hypotonia, lumbar lordosis, and delayed development were noted. At age 11, neuropsychological testing identified mild intellectual disability. From age 15, she experienced episodes of tonic/tonic–clonic seizures, but epileptic activity was absent in repeated EEGs. Lamotrigine treatment was partly effective. She continues to report muscular pain, fatigue, and anxiety. Recently, hyperprolactinemia was found (720 mlU L$^{-1}$ (ref: 102–496 mlU L$^{-1}$)). An MRI of the pituitary, however, did not disclose pituitary adenomas. She lives in her own home, but requires frequent supervision and help. For neuromuscular evaluation of all patients, including cerebral MRI with white matter changes, see Tables 1 and 2.

*Patient 1–2* is the younger brother of 1–1. Pregnancy and birth were uneventful. Delayed psychomotor development and lumbar lordosis was noted early alongside eating difficulties and low body weight. In childhood, he displayed left lower extremity weakness, hyperreflexia, and limping. Muscle biopsy showed unspecific changes of mitochondrial aggregation and muscle fiber caliber variation. Mild intellectual disability was diagnosed by neuropsychological testing in teenage years.

*Patient 1–3* is the younger brother of 1–1. Pregnancy and birth were unremarkable. He showed delayed early psychomotor development with generalized hypotonia and toe-walking. Hyperreflexia was noted in the lower extremities from age 5. Eating difficulties, nausea, abdominal pain, and low body weight were present from childhood. Neuropsychological testing at 5 years showed mild intellectual disability. Profound muscular pain and stiffness are reported after physical exercise. Mild bilateral high-frequency sensorineural hearing loss was found at age 12.

*Patient 1–4* is the youngest brother of 1–1. Pregnancy and birth were unremarkable. Eating difficulties and low body weight were observed in early years, with weight and length at the 2.5 percentile. Delayed psychomotor development, in-toeing, lumbar lordosis, generalized muscular hypotonia, and strabismus were noted. At age 7, he experienced a cerebral episode with coma and tetraplegia preceded directly by repeated vomiting, and a few days earlier by high fever. Serum measurements showed a metabolic acidosis with S-lactate 6, normal S-potassium (measured after intravenous infusion), and elevated S-creatin kinase (738) at hospital admission. Cerebral MRI revealed multiple lesions in both hemispheres periventricularly, subcortically, in the pons and the basal ganglia, in both gray and white matter. Acute disseminated encephalomyelitis was discussed, and steroid treatment was given. He regained consciousness over the following few days. Re-evaluation of MRI findings concluded with an "inflammatory perivascular reaction". MRI lesions were normalized a few weeks later, except for a white matter lesion by

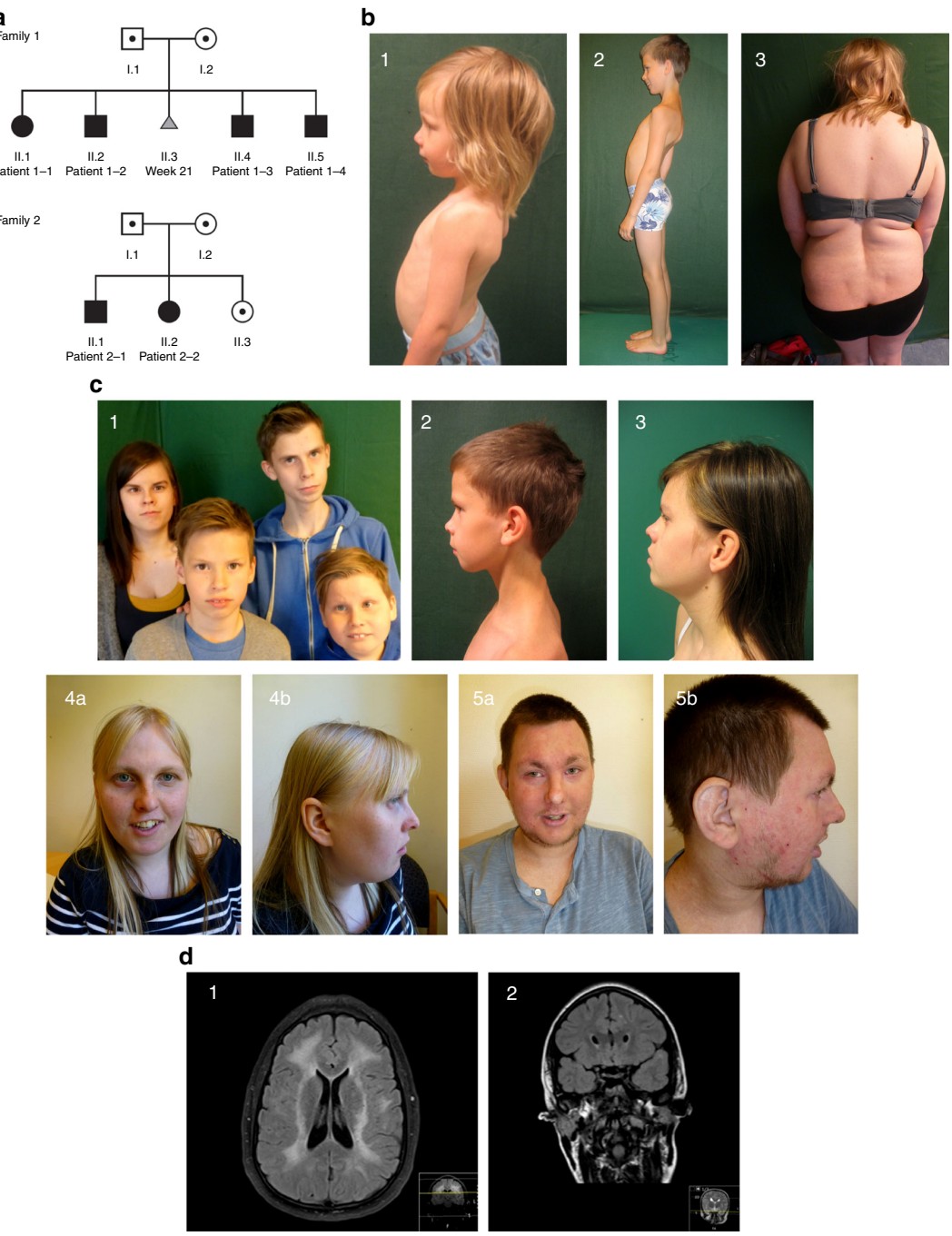

**Fig. 1** Clinical features and pedigrees of AIMS patients. **a** Pedigrees of both affected families. Black defines patients homozygous for the *ABCC9* c.1320 + 1 G < A mutation. Black dots indicate individuals heterozygous for the *ABCC9* variant. Gray triangle represents a fetus affected by probable thanatophoric dysplasia and terminated pregnancy. **b** Musculoskeletal features in AIMS patients. (1) Lumbar lordosis in patient 1–4 at age 4; (2) lumbar lordosis, thin habitus in patient 1–2 at age 10; (3) thoracolumbar scoliosis in patient 2-2 at age 28. **c** Facial features with prominent orbital ridges, hypotelorism, thin upper lip, flat midface in several of the patients. (1) Family 1. Upper left: patient 1–1 at age 20. Upper right: patient 1–2 at age 16. Lower left: patient 1–3 at age 11. Lower right: patient 1–4 at age 10; (2) profile of patient 1–2 at age 10; (3) profile of patient 1–1 at age 15; (4) patient 2–2 at age 28; (5) patient 2–1 at age 32. **d** Cerebral MRI of AIMS patients. (1) Magnetic resonance imaging (MRI) of the brain of patient 2–1. T2-weighted fluid-attenuated inversion recovery (FLAIR), coronal section shows widespread white matter hyperintensities. (2) MRI of the brain of patient 1–1. T2-weighted FLAIR, axial section shows juxtacortical white matter hyperintensities

the anterior horn of the left ventricle. He regained the same psychomotor level as before. Weight increased after that episode and is currently > 97.5 percentile. Neuropsychological testing at age 9 resulted in a diagnosis of mild intellectual disability. Obstructive sleep apnea, probably related to being overweight, a large tongue and hanging soft palate were diagnosed at age 11, and operative treatment was recommended. Mild bilateral high-frequency hearing loss was found at age 11.

*Individual II.3* was a female fetus, conceived inbetween patients 1–2 and 1–3. The parents elected to terminate pregnancy in

**Table 1 General clinical features and dysmorphology of AIMS patients**

| Patient | 1-1 | 1-2 | 1-3 | 1-4 | 2-1 | 2-2 |
|---|---|---|---|---|---|---|
| Age/sex | 21/female | 17/male | 13/male | 11/male | 33/male | 29/female |
| Cognitive function | Mild ID | Mild ID | Mild ID | Mild ID | Mild ID | Moderate ID |
| Hypotonia in childhood | + | + | + | + | + | + |
| Fatigability | + | + | + | + | + | + |
| Hearing | Very mild sensorineural hearing loss | N | High-freq mild hearing loss bilat | High-freq mild hearing loss bilat | N | N |
| Sleep aponoea | NA | NA | NA | + | + | NA |
| Psychiatry | Anxiety | Anxiety | NA | NA | Anxiety | Anxiety |
| Skeletal | Lumbal lordosis | Lumbal lordosis | Lumbal lordosis | Lumbal lordosis | Normal | Congenital hip dislocation Scoliosis Elbow extension deficit |
| Growth | L:2.5-10p W:2.5 -10p OFC:10p | L:10p W: < 2.5p OFC:2.5-10p | L:25p W: < 2.5p OFC:10p | L:10p W:97,5p OFC:50p | L: < 2.5p (162 cm) OFC:25p W: > 97.5p | L: < 2.5p (147 cm) OFC < 2.5p W: > 97.5p |
| Skin | Marmoration | Marmoration | Marmoration | Marmoration | Marmoration Depigmented patches Atopic dermatitis | Marmoration Depigmented patches Atopic dermatitis Telangiectasia left cheek 1 cafe au lait patch |
| Dysmorphology | | | | | | |
| Prominent supraorbital ridges | + | + | + | + | + | + |
| Hypotelorism | − | + | − | + | + | − |
| Broad nasal tip | + | + | − | − | + | + |
| Dental malocclusion | + | + | + | + | − | − |
| Flat face | + | + | + | + | + | − |
| Thin upper lip vermillion | + | + | + | + | + | + |
| Macrodontia upper central incisors | − | + | +, widely spaced | +, widely spaced | − | + |
| Food intolerance/ allergies | Milk protein intolerance | Lactose intolerance | N | Milk protein intolerance | Food allergies, anaphylaxia | Food allergies |

*ID* intellectual disability, *L* length, *N* normal, *NA* not assessed, *OFC* occipitofrontal circumference, *W* weight

pregnancy week 21, due to suspected thanatophoric dysplasia with micromelia and a narrow thorax.

**Family 2.** *Patient 2–1* is the first child of healthy parents. Pregnancy and birth were unremarkable. Delayed psychomotor development and muscular hypotonia were noted in toddler years. Due to similar findings in him and his sister, with a few depigmented skin patches and white matter lesions on cerebral computed tomography scan, a diagnosis of tuberous sclerosis (TS) was suggested, but later abandoned. The MRI pattern is not the same as in TS. Neither sibling has experienced epileptic seizures. Feeding was difficult in childhood, with low body weight, but he gained weight in adulthood, and is now overweight, with short stature. Severe atopic eczema is currently managed by cyclosporine, and he has multiple food allergies.

Recent neuropsychological testing led to a diagnosis of mild intellectual disability. He has been medicated for anxiety with a selective serotonin reuptake inhibitor for the last few years. He reports frequent dizziness attacks. Cardiac ultrasound at age 33 revealed biventricular systolic dysfunction, moderate left ventricle dysfunction, left ventricle ejection fraction (EF) of 35–40%, and raised NT-proBNP. Left ventricle diameter was within the normal range, although, cardiac MRI shows left ventricle dilatation, without obvious pathology of the myocardium—compatible with early-stage dilated cardiomyopathy. Treatment with an ACE-

antagonist and a beta-blocker was started. Hypertension was diagnosed before treatment initiation. A cardiac ultrasound in teenage years was normal. Cerebral MRIs at ages 16 and 33 showed widespread confluent white matter signal changes, described as similar to leucodystrophy. Lesions have increased significantly since the first investigation. MR angiography is normal, with normal cerebral vasculature calibers. Point lesions are found in the basal ganglia, pons, and white matter, representing possible mineral depositions. He lives in a sheltered home with daily help and supervision. He attends a sheltered work program, but is easily exhausted, and needs rest during the day. He is cheerful and social. Muscle strength is normal.

*Patient 2–2* is the younger sister of 2–1. Pregnancy and birth were unremarkable. She was treated for congenital hip dysplasia. Psychomotor development was delayed. Cerebral MRI in childhood showed periventricular white matter changes, and a diagnosis of tuberous sclerosis was considered (see above). Hypotonia, reduced muscle strength, and exhaustibility are reported since childhood. She has short stature and microcephaly. She had a thin build in childhood, but is now overweight. Recent neuropsychological testing indicates mild-to-moderate intellectual disability. She lives in her own flat with assistance. She has anxiety and has been followed by a local outpatient psychiatry service. Cardiac ultrasound at age 29 revealed biventricular systolic dysfunction, mildly reduced left ventricle ejection fraction (EF) of 48%, and a hypokinetic left ventricle with normal

**Table 2 Neurological, electrophysiological, and cardiac features of AIMS patients**

| Patient | 1-1 | 1-2 | 1-3 | 1-4 | 2-1 | 2-2 |
|---|---|---|---|---|---|---|
| *Neurological examination* | | | | | | |
| Cranial nerves | Nystagmus Convergent strabismus | Intermittent heterotropia | N | Bilateral intermittent esotropia | N | N |
| Muscle strength[a] | | | | | | |
| Hand grip | N | 4 | 4+ | 4 | N | 4 |
| Truncal muscle weakness | − | + | − | − | − | + |
| Proximal upper limb | N | N | N | 4 | N | 4 |
| Proximal lower limb | 4 | 4 | N | 4 | N | 4 |
| Achilles contractures | − | + | + | + | − | + (unilateral) |
| Hyperreflexia | Patella: brisk (+++) Achilles: subclonus | Brachioradialis/ triceps/achilles brisk (+++) | N | N | Patella, achilles: subclonus | Patella, achilles: subclonus |
| Balance (MiniBESTest) | Moderate–severe difficulties Score 16/28 | Moderate difficulties Score 20/28 | Moderate difficulties Score 18/28 | Moderate difficulties Score 21/28. | Moderate difficulties Score 18/28 | Moderate–severe difficulties Score 14/28 |
| 6-min walk test | Reduced (mean 512.5 m) Ref: 765 m | Reduced (mean 485.2 m) Ref: 725.8 m | Reduced (mean 467.5 m) Ref: 697.8 m | Reduced (mean 428.5 m) Ref: 672.8 m | Reduced (mean 507.5 m) Ref: 625 m | Reduced (mean 418.5 m) Ref: 668.7 m |
| *Electrophysiology* | | | | | | |
| EMG | N | N | N | N | N (slight polyfasia) | N |
| NCV | N | N | N | N | N | N |
| Repetitive nerve stimulation | N | N | N | N | N | N |
| EEG | N | NA | NA | N | NA | N |
| Muscle biopsy | NA | Caliber changes, mitochondrial aggregations. | NA | NA | NA | NA |
| Cerebral MRI | Small subcortical frontal hyperintensities MRS: N | Discrete periventricular white matter hyperintensities (posterior horns) | White matter hyperintensities, centrum semiovale | Periventricular white matter hyperintensities anterior horn of left lateral ventricle | Widespread white matter hyperintensities, periventricular and Centrum semiovale MRA: N | White matter periventricular hyperintensities MRA: N |
| *Cardiac examination* | | | | | | |
| Cardiac ultrasound | N | N | N | N | Biventricular systolic dysfunction, cardiac failure, dilated ventricles (cardiac MRI) | Mild biventricular systolic dysfunction |
| Blood pressure (mmHg) | 122/83 | 116/74 | 106/78 | 117/69 | 138/98 | 118/81 |

*EEG* electroencephalogram, *EMG* electromyography, *miniBESTest* mini balance evaluation system test, *MRA* magnetic resonance angiography, *MRI* magnetic resonance imaging, *MRS* magnetic resonance spectroscopy, *NCV* nerve conduction velocity, *N* normal, *NA* not assessed, *Ref* reference distance
[a]Muscle strength grading (0–5): 0 = paresis 4 = 50% strength reduction, 4 + = 25% strength reduction 5 = normal

diameter. ProBNP is normal and there are no clinical signs of cardiac failure at this time. Cerebral MRI performed recently shows an increase in periventricular white matter signal changes compared with the last MRI at age 13. MR angiography of cerebral vessels is normal.

*Individual II-2* is the mother of patients 2–1 and 2–2. Recent cardiac ultrasound, ECG and bicycle stress echocardiogram were normal at age 51.

*Individual II-3* is the younger sister of patients 2–1 and 2–2. She is healthy, and is the mother of two healthy children. Recent cardiac ultrasound, ECG, and bicycle stress echocardiogram were normal at age 25.

**Identification of a splice site mutation in ABCC9.** We performed sequencing of one affected individual from each family (patients 1–2 and 2–1), plus their respective healthy parents, using a targeted panel of > 4800 disease-associated genes. The samples were analyzed for recessive variants (homozygous and compound heterozygous) and non-Mendelian inheritance. The two trios were analyzed independently, and from a total of ~10,000 variants in each of the families, the homozygous variant *ABCC9* c.1320 + 1 G > A was the only remaining causal candidate after filtering against quality, region of interest, coding effect, minor allele frequency, and manual review of ~20 variants in each of the two families. Next-generation sequencing (NGS) and subsequent Sanger sequencing revealed that all six patients were homozygous for *ABCC9* c.1320 + 1 G > A. Analysis of DNA from the four parents and one unaffected sibling from family 2 showed that they were each heterozygous for the same mutation.

To exclude distant relatedness between the two families and thus exclude the possibility of the presence of more shared rare variants with an effect on protein function, we performed whole-genome sequencing on one affected individual from each family (patients 1–2 and 2–1), and determined a kinship coefficient of 0.0403. A kinship coefficient of ~0.05 is expected for unrelated samples, confirming non-relatedness between our families.

Since the original NGS analysis was performed on a gene panel, we analyzed the WGS data sets for candidate causal recessive and dominant variants. The focus of the analysis was the identification of shared variants or different variants in shared genes with possible damaging, but not identical variants leading to the same clinical phenotype, among the two cases. After the initial analysis, 67 shared variants were identified, of which 66 were heterozygous and 1 was homozygous (Supplementary Table 2). The single homozygous variant remaining after the filtering was the above variant in $ABCC9$:NC_000012.11:g.22063090 C > T; NM_020297.2:c.1320 + 1 G > A (Supplementary Fig. 2). This variant lies in a shared homozygous block of 3.8 MB (chr12:18.326.590-22.176.010 hg19). Notably, gene interactions with the homozygous $ABCC9$ c.1320 + 1 G > A variant cannot be excluded as participating in the syndrome. A list of all shared variants including allele frequencies is provided in Supplementary Table 2. For a dominant mode of inheritance, we observed that, in 842 genes, one or more variants pass the filtering criteria in both cases. Again, the $ABCC9$ variant was the only variant seen in homozygous state in both samples (Supplementary Fig. 3). Compound heterozygous gene candidate analysis identified seven genes with two or more variants shared between the two cases. None of these genes are likely to contribute to the phenotype (Supplementary Fig. 4). Thus, after gene panel and WGS, we identified the homozygous $ABCC9$ c.1320 + 1 G > A variant as the most likely causal variant. The variant is reported at very low allele frequency in the heterozygous state in the European population (Finnish: 3/6586; European: (Non-Finnish) 4/66386, Exac, June 2018). According to gnomAD, the variant is reported at surprisingly high frequency in heterozygous state in the Finnish population with an allele frequency of 0.0007 (18/24850). It is less common in non-Finnish Europeans (5/128232, allele frequency 0.00004) and absent in Asian or African populations (May 2019). Considering the probable Finnish ancestry of all patients, the syndrome might be more prevalent in the Finnish population than others. The homozygous state is absent in gnomAD.

The $ABCC9$ c.1320 + 1 G > A variant is predicted to disrupt a splice donor site of exon 8 (Fig. 2a). To study the effect of the mutation, we performed cDNA analysis on fibroblasts from members of both families (Fig. 2b). Sequence analysis of the homozygous cDNA samples revealed that the sequence variant caused an in-frame deletion of exon 8 (r.1165_1320del) in SUR2 cDNA, and predicts a 52 amino acid (p.Ala389_Gln440del) deletion within the SUR2 protein. Thus, all patients are homozygous for a splice variant in $ABCC9$ which results in an in-frame deletion.

**The effect of exon 8 deletion on $K_{ATP}$ channel function**. Deletion of exon 8 is predicted to disrupt multiple transmembrane helices in SUR2 and thus have a profound effect on structure and function (Fig. 3a). We deleted exon 8 in SUR2 cDNA (SUR2AΔ8) and performed expression analyses on Cosm6 cells transiently transfected with Kir6.2 alongside either Flag-tagged SUR2A-WT or SUR2AΔ8. Western blot of whole-cell lysates shows that exon 8 deletion results in a ~50% decrease in SUR2 protein expression (Fig. 3b). We used a radioactive rubidium ($^{86}Rb^+$) efflux assay to determine effects on $K_{ATP}$ channel function. Cells expressing SUR2A-WT with Kir6.2 exhibited robust rubidium efflux rates when $K_{ATP}$ channels were

activated by metabolic inhibition (Fig. 3c). In contrast, cells expressing Kir6.2/SUR2AΔ8 showed no rubidium efflux above the background levels observed in GFP-transfected cells. When Kir6.2 was co-expressed with a heteromeric 1:1 ratio of SUR2A-WT and SUR2AΔ8, $K_{ATP}$-dependent efflux rate was similar to WT rates, indicating that SUR2AΔ8 has no marked dominant-negative effect on functional $K_{ATP}$ expression. The effect of the exon 8 deletion in SUR2B was also determined and, again, no significant efflux was observed in cells expressing SUR2BΔ8 (Fig. 3d).

A complete absence of functional $K_{ATP}$ channels was observed in inside–out patch clamp recordings from cells co-transfected with Kir6.2/SUR2AΔ8, in contrast to robust expression in cells transfected with SUR2A-WT or a 1:1 mix of SUR2A-WT and SUR2AΔ8 (Fig. 3e, f). Therefore, homomeric deletion of exon 8 results in a significant decrease in protein expression and complete loss of $K_{ATP}$ channel function. The decrease in functional expression of SUR2Δ8 containing channels may arise due to either the absence of surface-expressed channels or the expression of nonfunctional channels, which requires more detailed study for elucidation. Co-expression of SUR2A-WT and SUR2AΔ8 did not affect channel regulation by ATP or pharmacological activation by pinacidil, suggesting that in the heterozygous context, the c.1320 + 1 G > A mutation is without significant effect (Fig. 3g; Supplementary Fig. 5).

**Fatigability and cardiac dysfunction in SUR2-STOP mice**. To model the effects of SUR2 LoF in vivo, we used a mouse line in which a frameshift mutation, resulting in a premature stop codon at position Y1148 (SUR2-STOP), was introduced using CRISPR/Cas9 (Fig. 4a). Inside–out patch clamp recordings from ventricular myocytes and aortic smooth muscle cells showed that functional $K_{ATP}$ channels were essentially absent in homozygous SUR2-STOP mice (Fig. 4b, c; Supplementary Fig. 6), thus the SUR2-STOP mice recapitulate the key functional channel consequences of the exon 8 deletion.

SUR2-STOP mice and WT littermate controls were evaluated on a multiple-trial inverted screen test to assess strength and fatigability. SUR2-STOP and WT mice performed comparably in the first trial, suggesting no significant initial deficits in strength per se. However, in subsequent repeated trials, SUR2-STOP mice exhibited clear decreases in performance, whereas WT mice performance remained high (Fig. 4d). Significant genotype effects were observed, as well as genotype x trial and genotype x session interactions (Supplementary Table 3). These findings indicate that SUR2-STOP mice exhibited significant performance deficits, dependent on the specific session and trial. The total time the mice remained inverted across the six trials was calculated, and a significant decrease in performance was observed in SUR2-STOP mice (Fig. 4e). Thus, global loss of SUR2 results in decreased physical performance, suggestive of increased fatigability, which may be related to the clinically observed myopathy.

Echocardiographic assessment revealed that left ventricle fractional shortening was significantly decreased in SUR2-STOP mice (Fig. 4f, g), consistent with the findings in older AIMS patients, and in previously reported SUR2-deficient mice[22]. A small, but statistically significant increase in left ventricular internal dimension in diastole (normalized to body length) was observed in SUR2-STOP mice, also mirroring the mild dilatation observed our eldest patient (patient 2–1), and the dilated cardiomyopathy previously associated with SUR2 mutations[18].

Blood pressure was significantly increased in SUR2-STOP mice (Supplementary Fig. 6C, D), consistent with the known role of SUR2-containing vascular smooth muscle $K_{ATP}$ channels in the regulation of vascular tone[23].

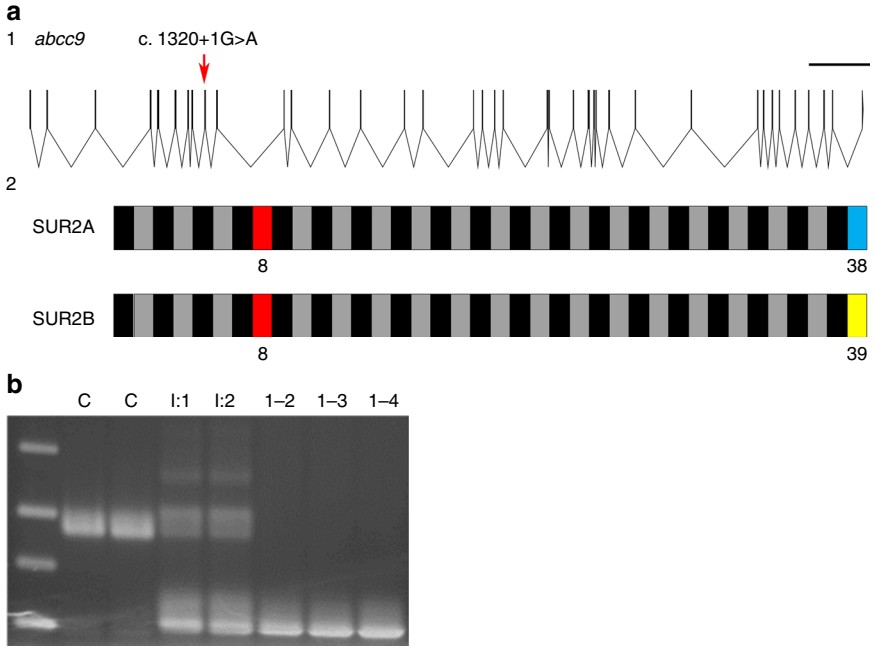

**Fig. 2** Molecular analysis in AIMS patients. **a** Genomic organization of the *ABCC9* gene: (1) basic genomic structure of *ABCC9* includes at least 39 potential exons, excluding untranslated regions (UTRs). The c.1320 + 1 G > A mutation predicted to disrupt the splice donor site of exon 8 is indicated by a red arrow. Scale bar, 5000 bp. (2) The mutation impacts both the SUR2A and SUR2B splice forms, which differ only in the last exon. Affected exon 8 in patients is marked in red. Odd-numbered exons are presented as black boxes, even-numbered exons as gray boxes. **b** Analysis of the effect of the mutation at the cDNA level in Family 1. Two control cDNA samples (indicated by a C) show the wild-type PCR product containing exon 8, parents (I:1, I:2) show heterozygosity for the wt and a lower band lacking exon 8, whereas patient cDNA (1–2, 1–3, 1–4) only yielded the lower band

**Absence of cognitive or behavioral defects in SUR2-STOP mice**. SUR2-STOP and WT mice were evaluated on a battery of cognitive and behavioral tests. During a 1-h locomotor activity test, SUR2-STOP mice displayed trends toward decreased total ambulations (whole-body movements) and vertical rearing frequency, and increased rest time, but there were no significant overall genotype effects (Fig. 5a–c). No differences were observed in the distance traveled in the center or peripheral zones of the test field, indices of emotionality in mice (Fig. 5d, e). In addition, neither significant effects were found in a battery of sensorimotor measures designed to assess balance, coordination, strength, and speed of movement (ledge test, platform test, pole test, inclined screen test, Fig. 5f–i) nor in tests of spatial learning and memory, evaluated using a Morris water maze (Fig. 5j–l). Finally, an elevated plus maze was used to assess anxiety-like behaviors, which involves quantifying the reluctance of mice to move from the "protected areas" of the enclosed arms to open arms. No significant differences were observed between genotypes in terms of distance traveled, time spent or entries made into the open arms, or total distance traveled throughout the entire maze (Fig. 5m–p). Collectively, the behavioral findings suggest that the SUR2-STOP mice do not exhibit marked deficits in learning and memory nor show any obvious anxiety-like behaviors.

**Decreased motility in SUR2-STOP zebrafish**. Assessment of a SUR2-STOP zebrafish model (Fig. 6a, b) also demonstrated clear phenotypic overlap with the clinical syndrome and mouse model. cDNA was analyzed by qPCR, to assess the effect of the S985 truncation, introduced via CRISPR/Cas9, on *abcc9* mRNA expression in SUR2-STOP zebrafish larvae. This revealed ~four-fold *abcc9* mRNA decrease in mutant fish compared with wild-type controls (Fig. 6c), consistent with aberrant mRNA being eliminated by nonsense mediated decay. Inside–out patch clamp recordings revealed the complete absence of functional $K_{ATP}$

channels in ventricular myocytes from SUR2-STOP zebrafish (Fig. 6d). Hence, the zebrafish model also recapitulates the key channel consequences of the exon 8 deletion.

Consistent with hypotelorism observed in multiple AIMS patients, significantly decreased interorbital distance, normalized to overall larval body length, was observed in SUR2-STOP zebrafish (Fig. 6d, e). In contrast, no significant difference was observed between the inter-eye distance in WT and SUR2-STOP mice. Visual inspection of SUR2-STOP zebrafish revealed no other striking dysmorphic features (Fig. 6b). We used a behavioral tracking system to quantify locomotor activity in zebrafish larvae (Fig. 6f). SUR2-STOP larvae displayed significantly decreased spontaneous total movement and decreased total swimming distance compared with control larvae (Fig. 6g, h). Despite shorter swimming distances, SUR2-STOP embryos move for a similar period of time as WT fish (Fig. 6i). Analysis of the duration of high-speed movements revealed a significant decrease in SUR2-STOP larvae (Fig. 6j). Notably, SUR2-STOP embryos hatched normally from their chorion, a process that also requires muscle contraction.

**Cardiac abnormalities in SUR2-STOP zebrafish**. We performed high-speed video imaging[24] of the hearts of wild-type and SUR2-STOP larvae to examine cardiac function (Fig. 7a; Supplementary Movies 1, 2). Analysis revealed systolic dysfunction—resembling the cardiac phenotype in older patients. Both fractional shortening (FS) and ejection fraction (EF) are significantly reduced in SUR2-STOP mutants (FS: 29%, EF: 25%) (Fig. 7b, c). Consequently, cardiac output is dramatically lower (28%) (Fig. 7d) due to equivalently reduced stroke volume (Supplementary Fig. 7A). Supplementary Fig. 8 illustrates the assessment of ventricular contractility via high-speed video microscopy in zebrafish embryos. Ventricular end-diastolic volume (VEDV) and end-systolic volume (VESV) were unchanged in SUR2-STOP larvae (Fig. 7e).

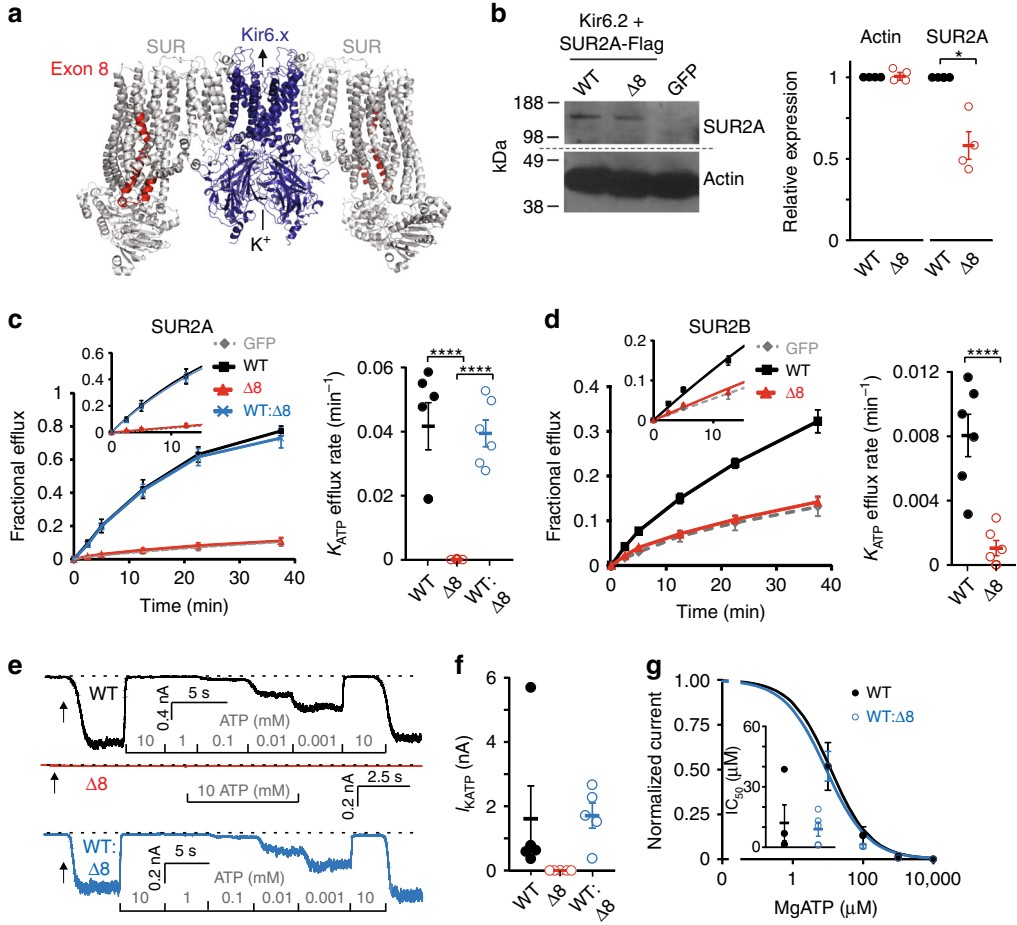

**Fig. 3** Exon 8 deletion results in $K_{ATP}$ channel loss-of-function. **a** $K_{ATP}$ channel structure showing pore-forming Kir6.x subunits (blue) associated with two of four SUR subunits (gray) (PDB 5WUA [doi 10.2210/pdb5WUA/pdb]), with the equivalent position of the SUR2 amino acids encoded by *ABCC9* exon 8 in red (Ala389-Gln440). **b** Western blot of whole-cell lysate from Cosm6 cells transiently transfected with GFP or Kir6.2 alongside either wild-type SUR2A-Flag (WT) or SUR2A-FlagΔ8. Left: Immunoblots using a primary antibody targeting the Flag-tag of SUR2A-FlagΔ8 (top) and actin control (bottom). Right: Normalized expression for actin and SUR2A-Flag from cells transfected with WT SUR2A-Flag or SUR2A-FlagΔ8. The data from four independent experiments, * denotes $p < 0.05$ according to Mann–Whitney $U$ Test. A representative example blot is included in the associated Source Data File. **c** Left: $^{86}Rb^+$ efflux from Cosm6 cells transfected with GFP alone, Kir6.2 alongside WT SUR2A, SUR2AΔ8, or a 1:1 ratio of WT SUR2A and SUR2AΔ8 (WT:Δ8). Inset: early time efflux/time data points used to derive efflux rate constants. Right: Efflux rate constants for cells transfected with WT SUR2A, SUR2AΔ8, and 1:1 ratio of WT:Δ8. The data from six replicates in three independent experiments, ****$p < 0.0001$ (Mann–Whitney $U$ Test). **d** Left: $^{86}Rb^+$ efflux experiment from Cosm6 cells transfected with GFP alone, with Kir6.2 alongside WT SUR2B or SUR2BΔ8. Right: Efflux rate constants for cells transfected with WT SUR2B or SUR2BΔ8. The data from six replicates in three independent experiments, ****$p < 0.0001$ (Mann–Whitney $U$ Test). **e** Example inside–out patch clamp recordings from Cosm6 cells transfected with Kir6.2 alongside SUR2A-WT (black), SUR2AΔ8 (red), or a 1:1 mix of SUR2A-WT and SUR2AΔ8 to mimic heterozygous expression (blue). The membrane potential was held at −50 mV in symmetrical KINT solutions, and ATP was applied as indicated. Arrows indicate the point of patch excision. **f** $K_{ATP}$ currents from excised patches. **g** ATP dose–response relationship for SUR2A-WT or 1:1 SUR2A-WT:SUR2AΔ8 channels. Inset: summary of ATP IC50 values. The data from individual experiments shown as dots alongside mean ± SEM. Source data are provided as a Source Data file

Blood flow velocity of WT and SUR2-STOP larvae was assessed by high-speed video imaging of the cardinal vein. An increased velocity of red blood cells in SUR2-STOP fish (Fig. 7f) can be associated with high blood pressure found in SUR2-STOP mice.

No cardiac abnormalities were observed in larvae heterozygous for the SUR2-STOP mutation (Supplementary Fig. 9).

Adult zebrafish hearts were analyzed after sectioning and H&E staining (Fig. 7g). For assessment of ventricular size, tissue sections revealing the largest chamber area were selected. In five out of six SUR2-STOP fish, ventricular area was strikingly enlarged with abnormal morphology compared with control siblings. The atrial area shows similar morphology (Fig. 7f; Supplementary Fig. 10). We stained cryosections of WT and

SUR2-STOP hearts with Acid Fuchsin Orange G (AFOG), which labels myocardium orange, collagen blue, and fibrin red. This revealed no visible fibrosis in SUR2-STOP hearts (Supplementary Fig. 11). TUNEL (TdT-mediated nick end labeling) analysis revealed very few apoptotic cells in WT hearts, while a sizable number of cells were TUNEL-positive in both cardiac chambers of SUR2-STOP fish (Fig. 7h). Myofiber structure in SUR2-STOP hearts is not different from WT as determined by immunohistochemistry staining for tropomyosin (Supplementary Fig. 12).

## Discussion
The major clinical features observed in our patients consist of delayed psychomotor development with intellectual disability,

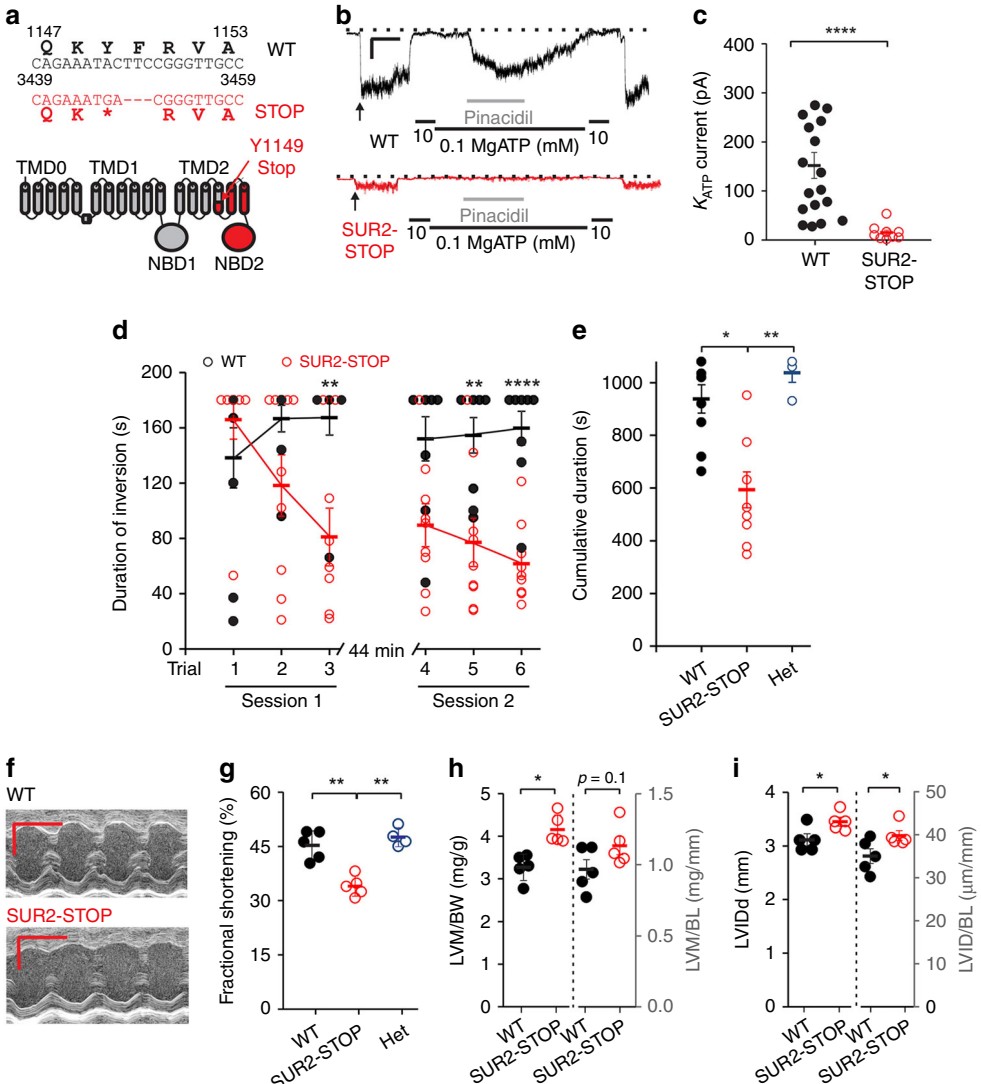

**Fig. 4** SUR2-STOP mice exhibit cardiac dysfunction and fatigability. **a** Top: The c.3446_3450delACTTCinsGA indel in *ABCC9* and consequent premature stop codon following K1148 (p.Y1149Stop). Bottom: schematic of SUR2 with the site of the introduced Y1149Stop mutation in TM15 indicated and the downstream region in red. **b** Example current traces from inside–out voltage clamp recordings from ventricular myocytes of WT (black) or SUR2-STOP (red) mice (−50 mV holding potential in the presence and absence of MgATP and pinacidil as indicated). Scale bar shows 5 s (*x*-axis) and 25 pA (*y*-axis). **c** $K_{ATP}$ channel current amplitudes from excised patches from mouse ventricular myocytes. The data shown from 18 patches for WT, and 10 patches for SUR2-STOP from ≥ 3 mice. ****$p < 0.0001$ (two-tailed *t* test). **d** Duration mice remained inverted during the multiple-trial inverted screen test. The data were analyzed using a repeated measures (rm) ANOVA model that contained one between-subjects variable (genotype) and two within-subjects variables (trials and sessions; see Supplementary Table 3 for summary statistics). The results from the rmANOVA revealed a significant genotype effect, as well as genotype x trial and genotype x session interactions. The data from nine WT and nine SUR2-STOP mice, *$p$-values for the pairwise comparisons exceeded Bonferroni correction ($p < 0.008$ [0.05/6]; *$p < 0.05$; **$p < 0.01$; ***$p < 0.001$). **e**) Cumulative inversion time, *$p < 0.05$ and **$p < 0.01$ according to one-way ANOVA and post hoc Tukey test. **f** Example M-mode echocardiography recordings from WT (top) and SUR2-STOP (bottom) mice. Scale bar shows 0.1 s (*x*-axis) and 1 mm (*y*-axis). **g** Ventricular fractional shortening measured from echocardiographic imaging (all echocardiographic data from five WT and five SUR2-STOP mice), **$p < 0.01$ (one-way ANOVA and post hoc Tukey test). **h** Left ventricular mass (LVM) as determined from echocardiography imaging normalized to body (LVM/BW) and body length (LVM/BL). *$p < 0.05$ (student's *t* test). **i** Left ventricular internal diameter in diastole as measured from echocardiographic imaging. *$p < 0.05$ (student's *t* test). The data from individual experiments shown as dots alongside mean ± SEM. Source data are provided as a Source Data file

anxiety, muscle weakness and fatigability, and some shared dysmorphic features. Cerebral MRI revealed white matter abnormalities in all subjects. Cardiac systolic dysfunction is found in the two oldest, possibly as an early stage of dilated cardiomyopathy. All patients were found to be homozygous for a previously unreported splice site mutation in *ABCC9* (c.1320 + 1 G > A), while unaffected parents are healthy heterozygous carriers of the variant (Fig. 2). No other shared recessive mutations were identified in the affected individuals using either next-generation gene panel analysis or whole-genome sequencing.

We show that this splice site mutation leads to the complete in-frame exclusion of exon 8 (SUR2Δ8) and consequent deletion of 52 amino acids within the TMD1 domain of the resultant SUR2 protein (Fig. 3). SUR2 is a regulatory subunit of $K_{ATP}$ channels expressed in various tissues, including striated and smooth muscle[3–5]. Deletion of exon 8 results in complete loss of

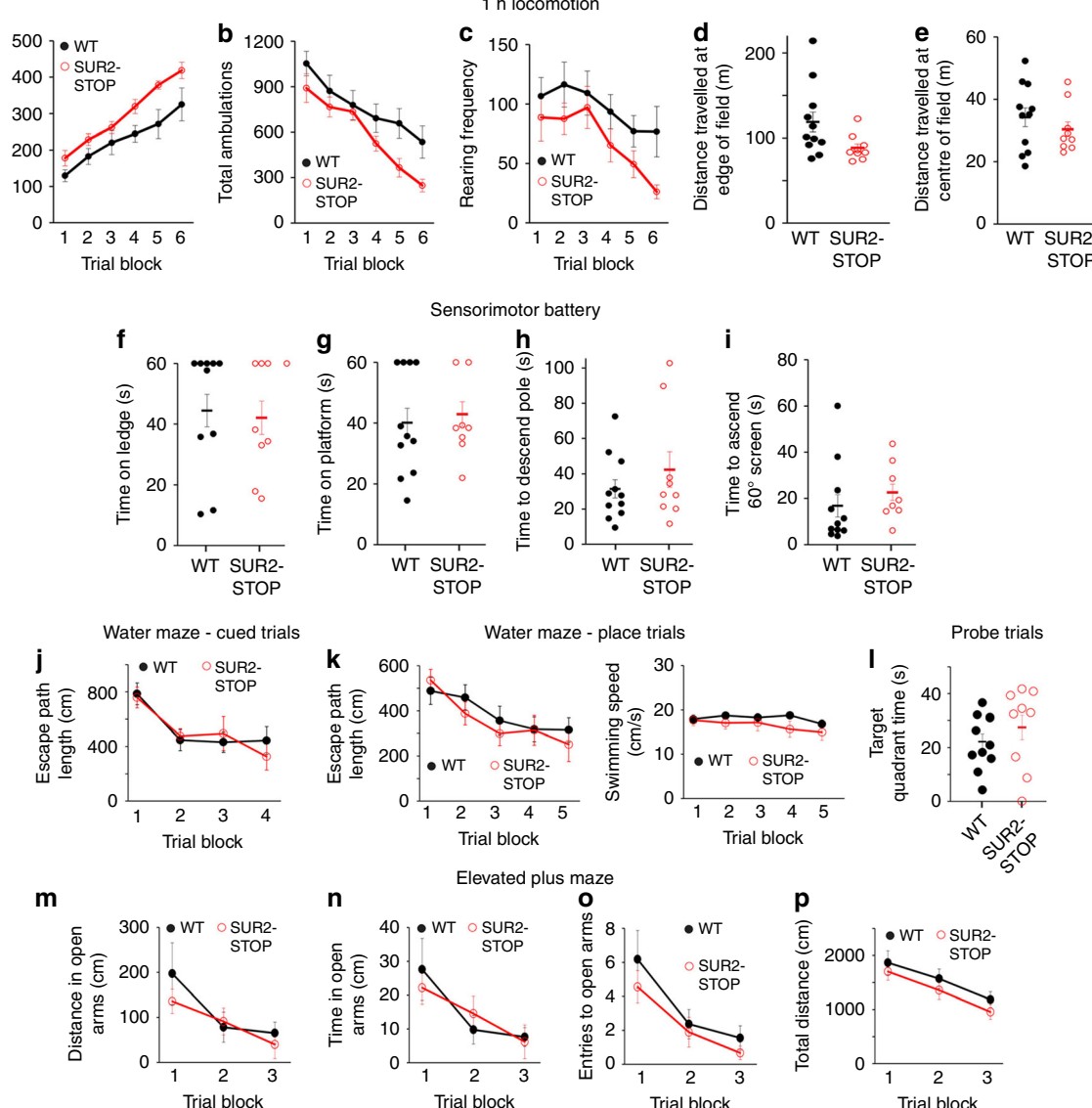

**Fig. 5** No major behavioral or cognitive deficits observed in SUR2-STOP mice. **a** Total time at rest **b** The total number of ambulations (whole-body movements), **c** the number of incidents of rearing, **d** distance traveled at edge of the observational field, and **e** distance traveled within center of observational field in 1 h locomotion observation, $n = 11$ for WT and 9 for SUR2-STOP for all behavioral tests. **f** Time mice spent on elevated ledge, **g** time mice remained on elevated circular platform, **h** time taken to descend during the pole test, and **i** time taken to ascend a wire-mesh screen maintained at 60º angle for WT and SUR2-STOP mice in sensorimotor battery. **j** The escape path length taken to find the platform by WT and SUR2-STOP mice in Morris Water Maze Cued trials. Note, no significant difference in swimming speed was observed between genotypes. **k** The mean escape path length (left) taken to find the platform, and average swimming speed (right) by WT and SUR2-STOP mice in Morris Water Maze Place trials. **l** The time spent within the target quadrant (the quadrant in which the platform had previously been positioned during Cued and Place trials) was measured. **m** Distance traveled within open arms, (**n**) time spent in open arms, **o** entries into open arms, and **p** total distance traveled in the Elevated Plus Maze test for WT and SUR2-STOP mice. The data were analyzed using a repeated measures (rm) ANOVA model that contained two between-subjects variable (genotype and sex) and one within-subjects variable (trial block). The data from individual experiments shown as dots alongside mean ± SEM. Source data are provided as a Source Data file

functional channel activity in recombinant SUR2/Kir6.2 $K_{ATP}$ channels, indicating that AIMS represents the human consequence of loss of SUR2 (Fig. 3). We show that key features of myopathy and cardiac dysfunction in AIMS are recapitulated in novel SUR2-STOP mouse (Figs. 4, 5) and zebrafish models (Figs. 6, 7). The animal models used in this study do not recapitulate the genetic defect identified in the AIMS patients, but were chosen as the functional effects of the frameshift mutations introduced into SUR2-STOP mice and fish mirror the functional effect of the SUR2Δ8 mutation. Future studies of CRISPR/Cas9 genome edited animal models in which human-disease-associated

AIMS mutations are introduced may provide further insights into the severity and variety of phenotypes arising from specific mutations.

How loss of SUR2-dependent $K_{ATP}$ channel function may result in the complex pathophysiology observed in AIMS is discussed below.

Facial features: Affected individuals from the two families share some similar facial features, including prominent supraorbital ridges, flat face, and thin upper lip vermilion, as well as macrodontia and/or widely spaced upper central incisors and dental malocclusion. Since the affected individuals are from two

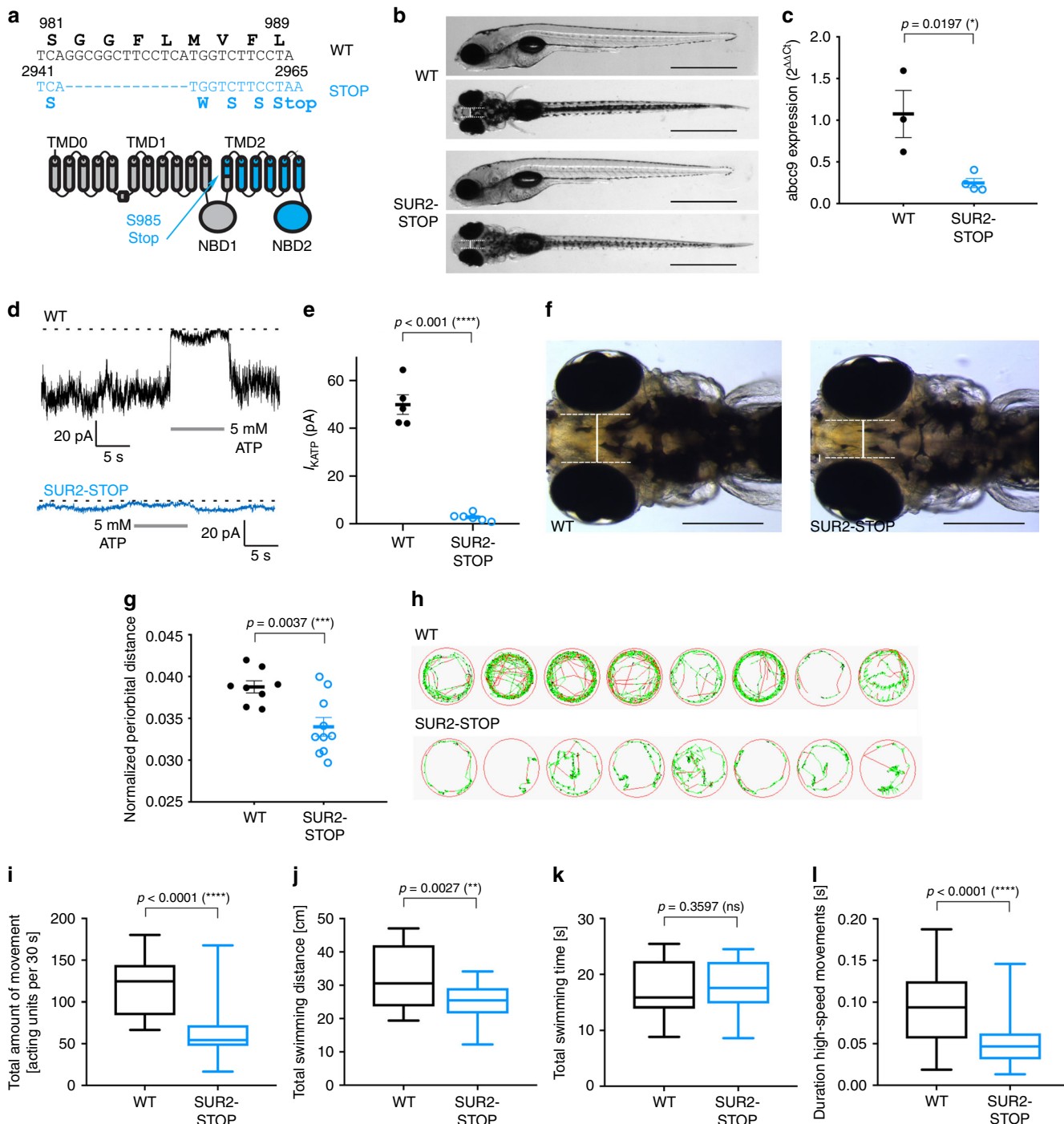

siblingships, more patient observations will be necessary in order to establish whether these facial features are consistent. Some of the individuals have hypotelorism, consistent with significantly decreased intraorbital distance found in SUR2-STOP zebrafish (Fig. 6f, g). This combination of facial features is markedly different from the acromegaloid facial features that characterize Cantú Syndrome and the associated conditions within the same spectrum: Acromegaloid Facial Features (AFF [MIM: 102150])[25] and hypertrichosis acromegaloid facial features disorder (HAFF)[26], which all arise from GoF mutations in *ABCC9*. None of our patients have hypertrichosis, unlike Cantú syndrome patients. Therefore, Cantú syndrome and AIMS, arising from opposing molecular mechanisms, are both dysmorphologically and mechanistically distinct.

Neuromuscular manifestations: All patients report fatigability, present with generalized hypotonia, and muscle strength is below normal in all but one. Lumbar lordosis is found in all but one, and scoliosis in one. The miniBEST test shows moderate to severe deficits in balance in all patients, and walking distance in the 6-min walk test is reduced in all individuals. S-CK, B-lactate, nerve conduction velocities, and electromyography, including repetitive nerve stimulation, are normal in all individuals, except for discrete myopathic discharges in patient 2–2. In skeletal muscle, $K_{ATP}$ channels are typically closed at rest but open in response to metabolic stress or fatigue[27]. Channel activation results in action potential shortening and stabilization of the resting membrane potential during the development of fatigue, which serves to reduce intracellular calcium, decrease resting

**Fig. 6** Hypotolerism and decreased locomotor behavior in SUR2-STOP zebrafish larvae. **a** The c.2944_2957del13 indel in *abcc9* and consequent frameshift premature stop codon following S984 (p.S985Stop) and schematic of SUR2 with the site of the introduced S985Stop mutation in TM12 indicated, downstream region in blue. **b** Representative images illustrating the morphology of 5 dpf wild-type and SUR2-STOP mutants as seen from a left lateral (top) and dorsal view (bottom). Scale bars, 1 mm. **c** Quantitative RT-PCR to assess *abcc9* expression in pools of 60 WT (three pools) and SUR2-STOP (four pools) embryos. **d** Representative current traces from inside–out patch clamp recordings from ventricular myocytes of WT (black; five patches) or SUR2-STOP (blue; six patches) zebrafish ($-50$ mV holding potential, ATP applied as indicated). **e** $K_{ATP}$ channel currents from excised patches from zebrafish ventricular myocytes. The data from five patches (WT), and six patches (SUR2-STOP) from $\geq 3$ zebrafish. **f, g** To assess hypotelorism, the distance between the convex tips of the eyes was measured and normalized to body length (WT, $n = 8$; HOM, $n = 10$). Scale bars, 200 μm. **h** Examples of movement traces shown in red, green, and black representing high-speed, intermediate, and slow movements, respectively. **i–l** Five-minute video recordings ($n = 62$ larvae per genotype) of 5 dpf homozygous SUR2-STOP fish and wild-type controls were analyzed for total amount of movement **i**, total swimming distance (TSD) **j**, total swimming duration **k**, and duration of high-speed movements **l** and compared with respective wild-type larvae. The data on the y-axis refer to the respective average value per 30-s (s) intervals. The data from four independent experiments (16 larvae per experiment) *$p \leq 0.05$; **$p \leq 0.01$; ***$p \leq 0.001$; ****$p \leq 0.0001$ (two-tailed unpaired Student's *t* test or Mann–Whitney *U* test). The black horizontal bar indicates the mean value for each condition. Sample size, WT, $n = 3$; SUR2-STOP, $n = 4$ in **c**; WT, $n = 5$; SUR2-STOP, $n = 5$ in **e**; WT, $n = 8$; SUR2-STOP, $n = 10$ in **g**; WT, $n = 62$; SUR2-STOP, $n = 62$ in **i–l**. The data from individual experiments shown as dots alongside mean ± SEM. Source data are provided as a Source Data file

tension and protect myocytes from damage. Therefore, loss of $K_{ATP}$ function might be expected to result in failure to recover from fatigue, myofiber degeneration, and excessive calcium influx, as reflected in the inverted screen tests in the mice. In addition, lower limb hyperreflexia was observed in the four oldest individuals, which might be caused by the white matter abnormalities and decreased inhibition or increased activation of upper motor neurons.

Although the expression pattern of Kir6.2 is not identical to that of SUR2, it is notable that previous studies of Kir6.2-null mice, as well as mice in which an internal deletion in SUR2 was engineered (resulting in loss of full length SUR2 but persistent expression of a mitochondria-limited short form), have reported that the consequent decrease of myocyte $K_{ATP}$ activity results in impaired exercise capacity and response, and myofiber damage[22,28]. Only relatively minor histological abnormalities were observed in a single-skeletal muscle biopsy from patient 1–2. This is consistent with previous reports which showed that SUR2 knockout mice only exhibit significant histological abnormalities when subjected to significant and chronic exercise[29]. Here, we show that novel mouse and zebrafish models, in which CRISPR/Cas9 was used to introduce frameshift mutations resulting in premature stop codons and nonfunctional subunits, also exhibit decreased performance. Specifically, SUR2-STOP mice showed a diminished ability to hang upside down during the multiple-trial inverted screen test, suggesting decreased strength resulting from increased fatigability[30,31] (Fig. 4d, e), while SUR2-STOP zebrafish show decreased total overall movement and swimming speed in tracking studies (Fig. 6i–l).

Intellectual disability and neurological abnormalities: Neuropsychological testing revealed mild-to-moderate intellectual disability in all affected individuals, and anxiety was also reported in several patients. It is not obvious how myocyte $K_{ATP}$ dysfunction could explain the intellectual disability or anxiety, and there are no reports of cognitive impairment in previous SUR2 mutant animal models. Neuronal $K_{ATP}$ channels are predominantly formed of Kir6.2 with SUR1 subunits, although transcripts for all $K_{ATP}$ channel subunits have been identified in various neuronal populations[32] and SUR2 is reportedly expressed in both central and peripheral neurons[33–35], where it has been implicated in hippocampal sclerosis in aging and amyotrophic lateral sclerosis[33,36–38].

However, recent data demonstrate that SUR2-containing $K_{ATP}$ channels play a critical role in regulation of cerebral vascular architecture[16]. In Cantú syndrome, MRI reveals diffusely dilated and tortuous cerebral blood vessels and white matter changes as multiple T2 subcortical or scattered hyperintensities. Transient white matter changes suggestive of a reversible posterior encephalopathic syndrome are reported in one patient. Several Cantú syndrome patients have migraine, and some have epilepsy. Developmental delay is common, although seeming to improve with age[16]. Since $K_{ATP}$ channel GoF results in chronic vasodilation and altered neuro–vascular coupling[16], it is conceivable that $K_{ATP}$ loss of function may impact the cerebral vasculature in a way that results in impaired dynamic coupling of blood flow to match neuronal metabolic demand.

Patient 1-1 has a diagnosis of epilepsy with episodes of unconsciousness and tonic-clonic seizures, but no definitive epileptic activity has been demonstrated on EEGs. Patient 1-4 experienced an episode of coma with transient widespread white and gray matter abnormalities, which is left unexplained. Possibly, these incidents could be caused by focal or generalized circulatory alterations. Interestingly, white matter hyperintensities are observed in both Cantú syndrome and AIMS. In both cases, they could result from ischemic events due to dysregulated cerebral blood flow, although the cognitive phenotype seems to be more definite in AIMS than in Cantú syndrome. Tests for cognitive deficits and anxiety did not reveal significant behavioral impairments in SUR2-STOP mice (Fig. 5). Additional testing for specific deficits (such as an assessment of working memory, fear conditioning, or other nonspatial forms of learning and memory) would be necessary to more fully characterize the behavioral phenotype of the SUR2-STOP mice. Notably, mild cognitive impairment, as seen in humans, can be difficult to recognize in animals[39]. Whole-genome sequencing and copy number analysis did not identify other causes of intellectual disability in our patients.

Cardiovascular manifestations: While the younger patients (aged 11–21) displayed no clear cardiovascular abnormalities, the two oldest patients display cardiac biventricular systolic dysfunction, with only slightly decreased ejection fractions. No clinical signs of heart failure were observed in patient 2–2, but decreased ejection fraction and heart failure was observed in her brother, patient 2–1. Cardiac MRI in patient 2–1 showed dilated ventricles, and together with other clinical findings this may suggest early stages of dilated cardiomyopathy. In a previous study by Bienengraber et al., two individuals with dilated cardiomyopathy (DCM) were found to have heterozygous, LoF, missense mutations in exon 38 of *ABCC9*, an exon only included in the SUR2A splice variant, and not in SUR2B[18]. In that case, $K_{ATP}$ LoF would be expected in cardiomyocytes and skeletal muscle but not smooth muscle, where SUR2B is expressed. The DCM patients reported in that study were older than the patients we report and displayed drastic reductions in ejection fraction (to

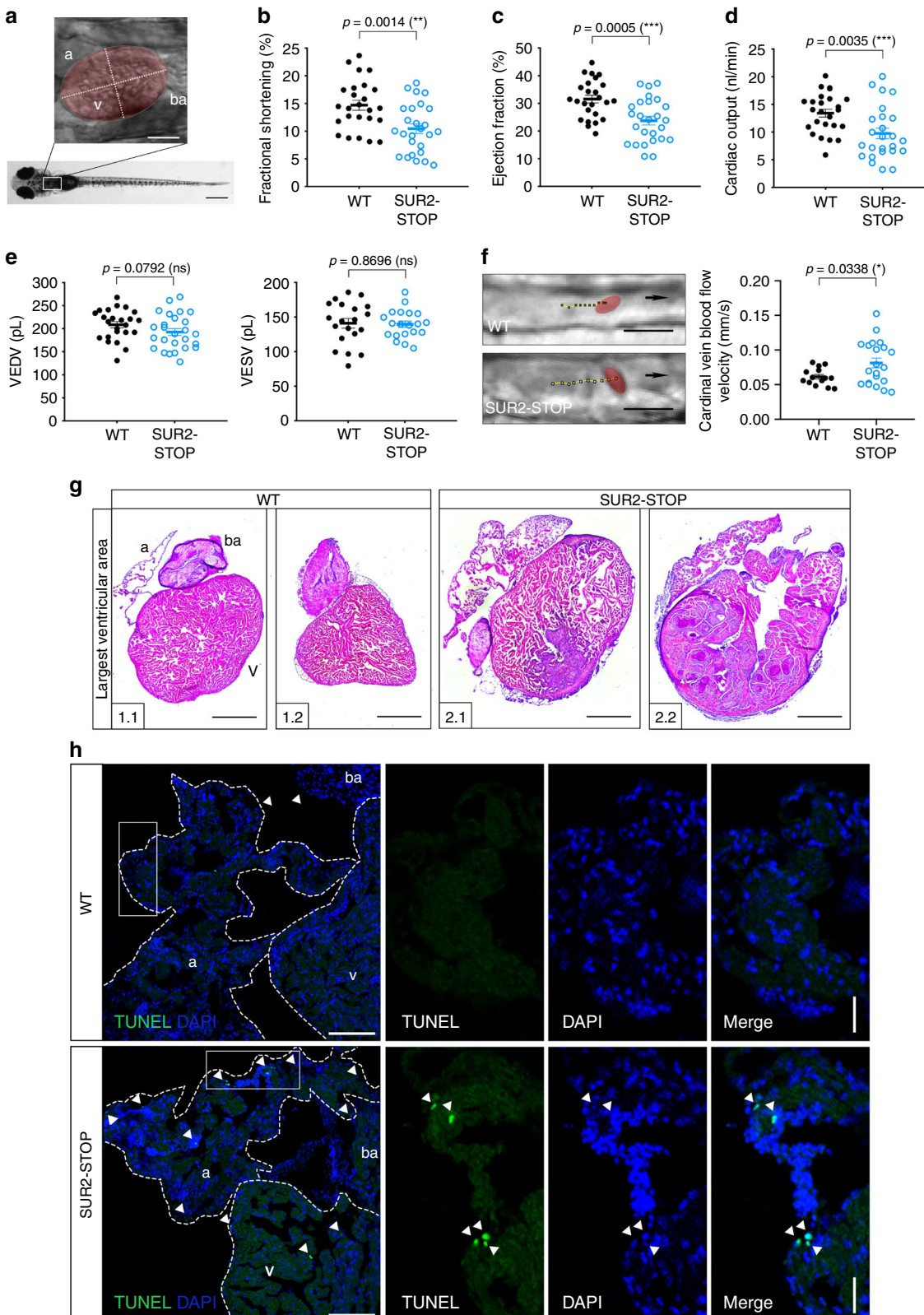

15–23%). This may suggest a progressive cardiomyopathy which will require longitudinal analysis in AIMS.

In previously reported mouse models, SUR2 knockout resulted in cardiac hypertrophy with LV dilation and decreased fractional shortening[22,40]. Here, we show that SUR2-STOP mice also display decreased fractional shortening and LV dilation (Fig. 4f, g), while SUR2-STOP zebrafish show marked cardiac enlargement with reduced ejection fraction and cardiac output (Fig. 7b–d), again consistent with the observed clinical phenotype in the older AIMS patients. Significant cardiomegaly, with elevated cardiac output, is observed in Cantú Syndrome. This cardiac remodeling may arise as a secondary compensation to counter the vasodilatory effect of $K_{ATP}$ GoF in vascular smooth muscle cells[41,42]. SUR2 LoF, in contrast, would be expected to increase vascular

**Fig. 7** Systolic dysfunction and enlarged heart size in SUR2-STOP zebrafish. **a** Box designates imaged area to assess cardiac function. The ventricular area of the heart is highlighted, with the long axis and short axis of the ventricle indicated by dashed lines. **b–e** Quantification of cardiac function using individual characteristic confocal sections from a time series of the embryonic cardiac cycle at 5 dpf. **f** Tracking of individual red blood cells (RBCs) measuring cardinal vein blood flow velocity. RBCs were tracked for ten frames using ImageJ (NIH) and the plugin MTrackJ[68]. One representative image of each genotype is shown. Black arrow indicates the direction of RBC movement. **g** Heart histology of adult SUR2-STOP mutants and respective siblings after H&E staining. Exemplary depiction of 2 WT and 2 SUR2-STOP hearts. For assessment of ventricular chamber size, tissue sections showing the largest ventricular area were selected. **h** TUNEL assay on adult hearts to detect apoptotic cells (white arrowheads) in WT and SUR2-STOP fish. Heart chambers are indicated by white dashed line. Nuclei are stained with DAPI. All experiments were performed comparing SUR2-STOP and its WT siblings. For all graphs, significance was determined by two-tailed unpaired Student's $t$ test or Mann–Whitney two-tailed U test. Asterisks indicate statistical significance (*$p \leq 0.05$; **$p \leq 0.01$; ***$p \leq 0.001$; ****$p \leq 0.0001$). The black horizontal bar indicates the mean value for each condition. Sample size, WT, $n = 20$; SUR2-STOP, $n = 20$ in **b–e**; WT, $n = 14$; SUR2-STOP, $n = 21$ in **f**; WT, $n = 6$; SUR2-STOP, $n = 6$ in **g** and **h**. Scale bars, 1 mm and 50 μm in **a**; 10 μm in **f**, 500 μm in **g**; 100 μm (overview) and 50 μm (close up) in **h**. All embryos analyzed originated from group matings of adult zebrafish. a atrium, ba bulbous arteriosus, v ventricle. The data from individual experiments shown as dots alongside mean ± SEM. Source data are provided as a Source Data file

contractility and blood pressure. However, blood pressure was normal in AIMS patients, except for elevated systemic pressures in the oldest patient. SUR2 and Kir6.1-null mice are hypertensive[43,44], which is again a feature of the SUR2-STOP mice, but in humans, effects on blood pressure may be more subtle and subject to tighter regulation through homeostatic feedback, or perhaps not yet manifesting in our still relatively young patients.

Other features: Mild hearing loss was found in 3/6 patients. This is not straightforward to explain, as there is no known function for SUR2 in auditory neurons. One explanation might be altered vascular tone and ischemic stress of these neurons. Strabismus and myopia were found in several individuals while one also displayed nystagmus. Sensory assessment was not performed in our animal models but should be the subject of future study. Feeding difficulties in childhood were reported in all individuals. $K_{ATP}$ channels are expressed in gastrointestinal smooth muscle[45], and thus LoF may affect GI contractility and motility. The two patients in family 2 have short stature and microcephaly. In Cantú syndrome, there is a range of skeletal manifestations. Our patients are not thoroughly evaluated radiologically, but do not display obvious skeletal changes, except for scoliosis, congenital hip dysplasia, and elbow extension deficits in patient 2–2. Lastly, patient 1–1 presents with hyperprolactinemia lacking pituitary adenomas. Interestingly, pituitary adenomas and elevated levels of prolactin are recently reported in a family with Cantú syndrome[46].

Disease progression: As discussed above, the two oldest patients exhibit decreased myocardial contractility, which may reflect the onset of a progressive cardiomyopathy, and hypertension is found in the oldest patient. It will be important to chronicle the long-term effects of $K_{ATP}$ LoF in all the patients to elucidate the natural history of AIMS, which may also be informed by longitudinal studies in the animal models presented here. White matter hyperintensities were found to have increased with age in the two oldest patients, where MRIs have been repeated.

Heterozygous carriers: Lack of phenotype in the healthy heterozygous carriers of the *ABCC9* c.1320 + 1 G > A variant is consistent with functional studies of recombinant $K_{ATP}$ channels reported here, which show that SUR2Δ8 subunits do not exert obvious dominant-negative effects in heterozygous expression with WT subunits. Indeed, essentially normal Rb fluxes in this case may reflect a rescue by WT subunits (Fig. 3c).

Possible additional molecular consequences: In addition to its canonical role as a regulatory subunit of plasmalemmal $K_{ATP}$ channels, short-form variants of SUR2 have been identified in the mitochondria and reported to form functional $K_{ATP}$ channels[6], although the role of these truncated channel complexes remain poorly understood. It is also conceivable that SUR2 proteins play

additional noncanonical (i.e., non-channel regulatory) roles, which may or may not be impaired by the exon 8 deletion. Importantly, the animal models employed here, in which premature stop codons were engineered into *ABCC9*, are not a bona fide recapitulation of the human genetics of the AIMS patients, and accordingly may not reiterate specific novel consequences of the truncated SUR2Δ8 protein.

Potential pharmacotherapy: In principle, $K_{ATP}$ channel LoF could be countered using $K_{ATP}$ channel activators, such as cromakalin, pinacidil, diazoxide or minoxidil, which are used clinically as vasodilatory agents[47]. In the case of the AIMS patients presented here, however, where the exon 8 deletion results in a complete loss of functional channel expression, these drugs would likely be ineffective. Interestingly, in each of the myocyte tissues in which SUR2 is expressed, $K_{ATP}$ forms part of an electrical "brake", where $K_{ATP}$ activation results in membrane potential hyperpolarization, decreased intracellular calcium, and promotes myocyte relaxation. Loss of $K_{ATP}$ function is therefore expected to increase intracellular calcium and modulating myocyte calcium handling might prove beneficial.

In the future, it is expected that AIMS patients will be identified with additional mutations in *ABCC9*. Such mutations may prove to be autosomal dominant or may exhibit only partial LoF, as is observed in the analogous pancreatic $K_{ATP}$ disorder, congenital hyperinsulism (CHI [MIM: 256450]), which arises from LoF mutations in *ABCC8*[48]. Depending on the mutational consequence, some CHI patients are responsive to treatment by the potassium channel opener diazoxide, and $K_{ATP}$-openers may indeed prove beneficial in some AIMS patients.

## Methods
**Patients**. Four siblings from Family 1 and two siblings from Family 2 (Fig. 1a, pedigrees) were initially referred for assessment of developmental delay and/or neuromuscular impairment, and were investigated in a clinical diagnostic setting. Written informed consent for Next-Generation-Sequencing in both a diagnostic and research setting was obtained from the patients' parents. The authors affirm that human research participants provided informed consent for publication of the images in Fig. 1b–d. Whole-genome sequencing on patients was approved by the Medical Ethical Committee of the University Medical Center Utrecht. All study participants, or their legal guardian, provided informed written consent prior to publication.

**Clinical evaluation by the Neuromuscular Centre team**. The clinical diagnostic evaluation included patient history, a structured neurological examination and physiotherapist examinations including the mini-Balance Evaluation System Test (miniBEST)[49] and 6-min walk test. The miniBEST is an assessment of dynamic balance and postural control. There are no normative values for children or people with intellectual disability, but the test was modified for our patients. The 6-min walking test- 6MWT[50,51] assesses distance walked over 6 min as a sub-maximal test of aerobic capacity/endurance. For adults, normative values can be calculated by an equation developed by Enright and Sherill (1998)[50]. The calculations are: men: $6MWD = (7.57 \times height\ cm) - (5.02 \times age) - (1.76 \times weight\ kg) - 309$. Women: $6MWD = (2.11 \times height\ cm) - (2.29 \times weight\ kg) - (5.78 \times age) + 667\ m)$. Normative values for children and adolescents are described by Geiger et al.[51].

Normative values are not given for intellectually disabled people. The neurological examination comprised the following: cranial nerve examination, motor function (strength, tempo, tone), deep tendon reflexes (biceps, brachioradialis, triceps, patella, achilles) and plantar reflexes, superficial skin sensation, vibration, proprioception, cerebellar function tests, and gait. In addition, the following supplementary electrophysiological investigations were performed: electromyography (EMG), nerve conduction studies (NCS), repetitive nerve stimulation. All patients had previously undergone cerebral magnetic resonance imaging (MRI). Blood samples were analyzed for serum levels of CK and lactate. In addition, all patients had audiometry, electrocardiograms (ECG), and electrocardiography done.

**Illumina TruSight One gene panel analysis.** DNA libraries for sequencing of 4813 genes with known associated clinical phenotypes were generated for patients 1–2 and 2–1 (Fig. 1) with their respective parents for trio analyses.

Genomic DNA was extracted from whole blood using MagAttract DNA midi 48 kit (Qiagen, katalognr 951356) on a Biorobot M48 (Qiagen) according to the manufacturer's protocol. Sequencing libraries were prepared from 50 ng of gDNA using the TruSight One Sequencing Panel (Illumina, San Diego, CA, USA) according to the manufacturer's instructions, Paired-end 150 bp sequencing was performed on a MiSeq (Illumina). Desktop Sequencer, targeting a mean region coverage depth > 100x and > 97% of the region at > 10 ×. Analysis of sequencing data was performed by applying a MiSeq Reporter Enrichment Workflow (v.2.4.60.8) including a Burrows–Wheeler Aligner (v.0.6.1-r104-tpx) and a Genome Analysis Toolkit (v.1.6-22-g3ec78bd), with the target region defined as the exome part with flanking 10 bp of 4813 disease-associated genes as defined by Illumina TruSight One Sequencing panel (http://www.illumina.com/products/trusight-one-sequencing-panel.html).

Cartagenia Bench Lab NGS (v.4) was used for variant annotation and prioritization, considering a recessive inheritance model using Cartagenia Inheritance analysis with the index patients defined as affected and the parents as unaffected. The focus of the analysis was to identify potentially damaging homozygous or compound heterozygous variants inherited in a recessive pattern from both the parents, or variants showing non-Mendelian inheritance, indicating a de novo event. Variants were filtered based on quality criteria (LowGQX < 10, LowVariantFreq < 0.20, LowGQ < 30.00, R8 > 8, SB > -10, LowDP < 20) and their presence in population frequency databases with a maximum minor allele frequency of 1 % (ExAC, 1000 genomes, dbSNP. Splice-site analysis was restricted to 8 intronic bp positions from the nearest intron–exon boundary and 3 exonic bp positions from the nearest intron–exon boundary. Findings in patients 1–2 and 2–1 were validated by Sanger sequencing. Sanger sequencing for the *ABCC9* sequence variant was performed for both healthy and affected family members.

**Whole-genome sequencing.** DNA libraries for Illumina sequencing were generated for patients 1–2 and 2–1 using standard protocols (Illumina) from 200 –1 μg of genomic DNA. The libraries were sequenced with paired-end (2 × 150 bp) runs using Illumina HiSeq X Ten sequencers with a target depth of 30 × base coverage. The Illumina data were processed with our in-house developed pipeline v 2.5.1 (https://github.com/UMCUGenetics/IAP) including Genome Analysis Toolkit (GATK) v3.4-46[52] according to the best practices guidelines[53]. Briefly, sequence reads were mapped against the human reference genome GRCh37 using Burrows–Wheeler Aligner v0.7.5a mapping tool[54]. Sequence reads were marked for duplicates using Sambamba v0.5.8[55] and recalibrated per sample using GATK BaseRecalibrator. Next, GATK Haplotypecaller was used to call SNPs and indels to create gVCF's. These gVCF's were genotyped with GATK. Variants are flagged as PASS only if they do not meet the following criteria: QD < 2.0, MQ < 40.0, FS > 60.0, HaplotypeScore > 13.0, MQRankSum < -12.5, ReadPosRankSum < -8.0, snpclusters > = 3 in 35 bp. For indels: QD < 2.0, FS > 200.0, ReadPosRankSum < -20.0. WGS statistics are provided in Supplementary Table 1.

Full analysis was performed on a merge of the reference coding regions obtained from Ensembl (v.75) (http://www.ensembl.org/) and UCSC (https://genome.ucsc.edu/) with ± 100 bp flanks. Cartagenia Bench Lab NGS v.4.3.5 was used for variant interpretation and prioritization. Variant analyses were made considering a recessive inheritance model using the Cartagenia cohort analysis tool. The focus of the analysis was to identify shared variants or shared genes with possible damaging, but not identical, variants, amongst the two cases. Lenient quality parameters were used to determine high quality variant calls (read depth > 4 reads, filter:pass, genotype quality > 98). Variants were filtered based on presence in population frequency databases with a maximum MAF of 1% (ExAC, 1000 genomes, dbSNP, and GoNL). Variants are categorized as LoF (frameshift, stopgain, startloss, stoploss), nonsynonymous, synonymous, and inframe variants. Furthermore, splice-site analysis was restricted to 8 intronic bp positions from the nearest intron–exon boundary and 3 exonic bp positions from the nearest intron–exon boundary. The Kinship coefficient was calculated from WGS data. Notably, due to multi-sample calling and noise present in the data, the kinship coefficient of unrelated samples is expected to be around 0.05. Hence, this is usually applied as proximate to calculate relatedness.

**cDNA analysis.** RNA was isolated from fibroblast cultures generated from skin biopsiesusing an RNeasy minikit (Qiagen) according to the manufacturers

protocol. cDNA was produced using SuperScript® VILO™ cDNA Synthesis Kit (Invitrogen). Two microliters of cDNA were used in a PCR reaction amplifying *ABCC9* exon 6–9 using primers as described in the Supplementary Methods. PCR products were subsequently verified by Sanger sequencing.

**Molecular biology and recombinant expression.** For recombinant protein expression and functional characterization, a 52 amino acid deletion (Ala386 to Gln437; Δ8) equivalent to the human exon 8 coding region was engineered into either a N-terminal Flag-tagged rat SUR2A construct (pcDNA_rSUR2A-Flag; GenBank accession No. D83598.1), a wild-type rat SUR2A construct (pCMV_SUR2A-WT; GenBank accession No. D83598.1) or a wild-type rat SUR2B construct (pcDNA_rSUR2B-WT; GenBank accession No. AF019628.1). The rat SUR2 sequence is 97% identical to the human SUR2 and was used to allow for comparison with an extensive number of previous studies of recombinant SUR2[13,14,56–58]. SUR2 cDNA (1 μg) was transiently transfected into Cosm6 cells[59] (RRID: CVCL_8561) using Fugene 6 (Roche) alongside wild-type Kir6.2 (0.6 μg; pcDNA3.1_mKir6.2; GenBank accession No. D50581.1). To mimic heterozygous expression, Kir6.2 was co-transfected with a 1:1 ratio of SUR2A-WT and SUR2AΔ8 (0.5 μg:0.5 μg). Western blot analysis and radioactive rubidium efflux assays were performed 48–72 h after transfection.

**Western blot analysis of recombinant SUR2.** Cosm6 cells were transfected with GFP alone, Kir6.2 with Flag-tagged SUR2A-WT or Kir6.2 with Flag-tagged SUR2AΔ8 (N-terminal Flag tag with amino acid sequence MDYKDDDDKGAP was inserted prior to the SUR2A start codon). After 48 h, cells were harvested in phosphate buffered saline (PBS; Gibco), pelleted, and lysed in 50 μl of PBS containing 1% Triton X-100 (Sigma) by mixing for 1 h at 4 ℃. Cell lysate was mixed with 2× SDS loading buffer, and was separated by electrophoresis using a precast 4–12% Bis-Tris Gel (Nupage). Gel contents were transferred to membranes using an iBlot 2 (Thermo-Fisher). Membranes were first incubated in a blocking solution (PBS/1% TWEEN 20/0.4% milk; PBSTM) for 1 h at room temperature (RT) with gentle shaking, before being washed (3× using PBS/TWEEN) and incubated in the primary horseradish peroxidase-conjugated anti-Flag antibody (Anti-Flag M2-HRP-conjugate Sigma # A8592; 1 in 5000 in PBSTM), or an anti-α actin antibody (clone C4 MilliporeSigma # MAB1501;1 in 3000 in PBSTM) for 1 h/RT. Anti-actin blots were washed (3× in PBSTM) and incubated with horseradish peroxidase-conjugated secondary antibody (Goat anti-mouse IgG antibody-HRP conjugate, Santa Cruz Biotechnology # sc-2005; 1 in 5000 in PBSTM) for 1 h at room temperature. A Pierce ECL Western Blotting Substrate kit was used for chemiluminescence detection; blots were analyzed and protein density quantified using ImageJ (NIH) and Excel (Microsoft). Actin and SUR2AΔ8 expression was normalized to values observed in WT SUR2A-Flag transfected cells in each independent experiment.

**Rubidium ($^{86}$Rb$^+$) efflux assay.** Cosm6 cells were plated and transfected in 12-well plates at sufficient density to allow for ~75% confluence on the day of experimenting. Cells were loaded with radioactive $^{86}$Rb$^+$ by incubation (> 6 h) with DMEM media spiked with 1 μCi ml$^{-1}$ $^{86}$RbCl (PerkinElmer) at 37 ℃/5% CO$_2$. After isotope loading, cells were washed in Ringer's solution which contained (in mM): 118 NaCl, 10 HEPES, 25 NaHCO$_3$, 4.7 KCl, 1.2 KH$_2$PO$_4$, 2.5 CaCl$_2$, and 1.2 MgSO$_4$, and was supplemented with 2.5 mg/ml oligomycin and 1 mM 2-deoxy-D-glucose to induce metabolic inhibition (MI), and incubated at room temperature for a further 10 min. Cells were then washed and Ringer's solution was added to each well, before being collected and replaced at the defined time points (2.5, 5, 12.5, 22.5, and 37.5 min). Cells were then lysed with 2% SDS to liberate remaining intracellular $^{86}$Rb$^+$ and sample radioactivity was determined by scintillation counting.

Cumulative $^{86}$Rb$^+$ efflux was calculated from the total counts from each time point. A time-dependent divergence from a mono-exponential efflux was observed, as noted previously[58], and thus rate constants were derived from exponential functions fit to early time points only (2.5–12.5 min). K$_{ATP}$-independent efflux rate constants (k$_1$) were obtained from GFP-only transfected cells using the equation:

$$\text{Efflux} = 1 - e^{-k_1 \cdot t}, \qquad (1)$$

The K$_{ATP}$-dependent efflux rate constant (k$_2$) was obtained from K$_{ATP}$-transfected cells using the equation:

$$\text{Efflux} = 1 - e^{((-k_1 \cdot t) + (-k_2 \cdot t))}, \qquad (2)$$

where k$_1$ was obtained from GFP-transfected cells (Eq. (1)), and the number of active channels was assumed to be proportional to k$_2$. Efflux-time data shown represent the mean ± SEM from at least three independent experiments with multiple internal replicates ($N \geq 3$, $n \geq 4$). The Mann–Whitney $U$ test was used to determine statistical significance at a $p$- value < 0.05.

**Patch clamp recording of recombinant channels.** Cosm6 cells were transfected with WT Kir6.2 (0.6 μg) and either WT SUR2A (1 μg), SUR2AΔ8 (1 μg), or a 1:1 ratio of WT and SUR2AΔ8 (0.5 μg: 0.5 μg), and 0.1 μg pcDNA3.1-eGFP to identify transfected cells. In all, 48–72 h after transfection, excised inside out patch clamp recordings were made. The membrane potential was voltage clamped at −50 mV.

Symmetrical KINT solution which contained (in mM): 140 KCl, 10 HEPES, 1 EGTA (pH 7.4 with KOH) was used for pipette and bath solutions. ATP and pinacidil were applied (in the presence of 0.5 mM free $Mg^{2+}$, calculated using CaBuf (Katholieke Universiteit Leuven) using a Dynaflow Resolve perfusion chip (Cellectricon). Experiments were performed at 20–22 °C. $K_{ATP}$ channel currents in the absence and presence of nucleotides and pinacidil were normalized to the basal current in KINT, and dose–response data were fit with a four-parameter Hill fit according to Eq. (3), using the Data Solver Function in Microsoft Excel, where the current in KINT = Imax = 1; Imin is the normalized minimum current observed in ATP; [X] refers to the concentration of ATP; $IC_{50}$ is the concentration of half-maximal inhibition; and H denotes the Hill coefficient.

$$Normalized\ current = Imin + (Imax - Imin)/(1 + ([X]/IC_{50})^H). \qquad (3)$$

**Generation of the SUR2 LoF mouse.** All mouse studies were performed in compliance with the standards for the care and use of animal subjects defined in the NIH Guide for the Care and Use of Laboratory Animals, and were reviewed and approved by the Washington University Institutional Animal Care and Use Committee. SUR2-STOP mice were generated using CRISPR/Cas9-mediated genome engineering[60]. In mice in which an attempt was made to generate a Cantú Syndrome-associated missense mutation, an artefactual indel mutation in *ABCC9* (c.3446_3450delACTTCinsGA) was identified in one founder individual (B6CBA F1/J background), which results in the introduction of a premature stop codon (p. Y1148Stop). This founder (termed heterozygous SUR2-STOP) was viable and fertile and was crossed with C57BL/6J mice to generate F1 SUR2$^{wt/STOP}$ mice. gDNA was isolated from F1 mouse tails, and PCR was used to verify the presence of the indel mutation in positive offspring. One positively identified SUR2$^{wt/STOP}$ F1 mouse was selected, and further crossed against C57BL/6J to generate F2 SUR2$^{wt/STOP}$, which were in turn intercrossed to generate F3 homozygous SUR2$^{STOP/STOP}$ (hereafter referred to as SUR2-STOP) and wild-type and heterozygous littermates, which were used for experiments.

**Electrophysiological recordings of isolated myocytes.** Ventricular myocytes were isolated from adult mice as previously described[61]. Inside–out patch clamp recordings were made in symmetrical pipette and bath $K_{INT}$ solutions which contained (in mM): 140 KCl, 10 HEPES, 1 EGTA (pH 7.4). Solution exchange was achieved using a Dynaflow Resolve perfusion chip (Fluicell). MgATP and pinacidil (both Sigma) were applied as indicated. Free $Mg^{2+}$ was maintained at 0.5 mM by supplementing ATP-containing solutions with $MgCl_2$, where necessary. Recording electrodes were formed from sodalime hematocrit glass (Kimble) and had a resistance of 1–1.4 MΩ when filled. Currents were recorded at −50 mV, sampled at 3 kHz, and filtered at 1 kHz using an Axopatch 700B and Digidata 1200 (Molecular Devices). $K_{ATP}$ current was calculated as the difference between the current in the absence of nucleotides and in the presence of a fully inhibiting concentration of 10 mM MgATP.

Vascular smooth muscle cells were acutely isolated form the descending thoracic aorta, as previously described[42]. Whole-cell patch clamp recordings were made in an initial $Na^+$ bath solution (which contained, in mM): NaCl 136, KCl 6, $CaCl_2$ 2, $MgCl_2$ 1, HEPES 10, and glucose 10 (pH 7.4) before switching to a K+ bath solution which contained (in mM) KCl 140, $CaCl_2$ 2, $MgCl_2$ 1, HEPES 10, and glucose 10 pH 7.4, in the absence and presence of pinacidil and glibenclamide as indicated. The pipette solution contained (in mM) potassium aspartate 110, KCl 30, NaCl 10, $MgCl_2$ 1, HEPES 10, $CaCl_2$ 0.5, $K_2HPO_4$ 4, and EGTA 5, with pH adjusted to 7.2 with KOH. The glibenclamide sensitive current density at a holding potential of −70 mV was taken as a measure of $K_{ATP}$ channel activity.

The data were analyzed using Clampfit (Molecular Devices) and Excel (Microsoft) and are presented as individual replicates in scatter plots, alongside mean ± SEM. The two-tailed $t$ test was used to determine statistical significance at a $p$-value < 0.05 using the RealStatistics add-in for Microsoft Excel.

**Mouse echocardiography.** Echocardiography was performed on adult (5-month-old) mice, as previously described[62]. Images were acquired using the Visual Sonics Vevo 770 Imaging System (Visual Sonics Inc., Toronto, Canada) and analyzed with Visual Sonics software. Mice were lightly anesthetized using Avertin and M-mode images of the parasternal long axis were obtained, from which measurements of fractional shortening, left ventricular mass, and left ventricular internal dimension in diastole (LVIDd) were calculated. Left ventricular mass and LVIDd were normalized to the mouse body weight and length (tip of nose to base of tail) for statistical comparisons. One-way ANOVA followed by post hoc Tuket test or Two-tailed $t$ tests were used to determine statistical significance using the RealStatistics add-in for Microsoft Excel, as described.

**Multiple-trial inverted screen test.** The procedure involved a modification of a previously published single-trial method that had been used to demonstrate impaired strength in a mutant mouse model of muscular dystrophy[31]. This methodology was adapted to evaluate deficits in strength related to decreases in physical endurance, which involved conducting multiple test trials administered in a single day. The apparatus was a wire-mesh screen (16 squares per 10 cm) that was elevated ~50 cm. A trial consisted of placing an adult (2–3-month-old) SUR2-

STOP mouse, heterozygote (Het) SUR2-STOP mouse, or a wild-type littermate control on the screen that was inclined to 60° with its head oriented down toward the base of the screen. After the mouse was stabilized on the screen, it was slowly inverted resulting in the mouse hanging upside down from the screen. Soft bedding was placed underneath the mouse to help protect it from injury. The time a mouse remained hanging upside down following inversion of the screen was recorded during an initial 3-min test trial. After the first trial, a mouse was placed into a holding cage for 5 min after which two more trials were conducted with 5 min being spent in the holding cage between trials. After completing the three trials, a mouse was placed back into the holding cage where it remained for 44 min, which was followed by the same 3 × 3-min test trial procedure described above. If a mouse remained hanging from the screen for the full 3 min duration, it was removed from the screen and assigned a value of 180 s. The time a mouse remained hanging upside from the screen (3-min maximum) was analyzed for each of the six trials as well as the total duration summed across all six trials.

**Behavioral and cognitive testing of SUR2-STOP mice.** Cognitive and other behavioral functions were assessed in adult (4–5-month-old) SUR2-STOP mice and wild-type littermates by conducting a battery of tests to evaluate ambulatory activity, emotionality, sensorimotor capabilities, learning and memory, and anxiety-like behaviors. The battery included the following tests (in order): 1-h locomotor activity; a series of seven simple sensorimotor measures; the Morris water maze, object recognition; and elevated plus maze, respectively. Full descriptions of these procedures are provided in the Supplementary Methods.

**Blood pressure recordings in SUR2-STOP mice.** Arterial blood pressures were measured in 12-month-old SUR2-STOP mice and WT littermates anesthetized with 1.5% isoflurane. The left carotid artery was catherized and a Millar pressure transducer advanced to the ascending aorta. Pressures were recorded using Powerlab data acquisition (ADInstruments), and mean arterial pressure (MAP) was calculated using LabChart 7 (ADInstruments).

**Zebrafish maintenance.** All zebrafish experiments were conducted under the guidelines of the animal welfare committee of the Royal Netherlands Academy of Arts and Sciences (KNAW) and were approved by the local ethics committee at the Royal Dutch Academy of Sciences (KNAW). Adult zebrafish (*Danio rerio*) were maintained at 28 °C and subjected to the 14-h light/10-h dark cycle. Embryos were produced by natural mating and kept in a humidified incubator at 28 °C[63]. All CrispR/Cas9 injections were performed in the wild-type strain Tübingen longfin.

**CRISPR/Cas9 design and embryo injections.** A target site within the abcc9 open-reading frame was chosen using CHOPCHOP (http://chopchop.cbu.uib.no/). sgRNA template was purchased at Integrated DNA Technologies (IDT) as standard desalted same-day oligos and synthesis was carried out using the Ambion MEGAscript T7 or SP6 kits (Ambion). Purification of the in vitro synthesized mRNA was achieved with the RNeasy Mini Kit (Qiagen)[64]. gRNA and Cas9-encoding mRNA were co-injected into one-cell stage zebrafish embryos. Each embryo was injected with 2 nl of solution containing ~12.5 ng μl$^{-1}$ of sgRNA and ~300 ng μl$^{-1}$ of Cas9 mRNA. Injected embryos were grown to adulthood to generate "founder" fish (F0), and screened for germline transmission of CRISPR-induced indel mutations using an adapted method for detecting simple sequence-length polymorphisms (SSLPs). Each putative founder adult fish was crossed with a wild-type adult fish (F1). Sequencing analysis confirmed successful introduction of 13n deletion and segregation in a Mendelian fashion. Homozygous fish (F2) were generated by inbreeding heterozygous mutant carriers.

**Quantitative RT-PCR.** Zebrafish larvae at 5 dpf were pooled in groups of 60 (three different groups for WT and four groups for homozygous mutants) and homogenized. RNA was purified using Trizol following standard procedures and cDNA synthesized using High-Capacity cDNA Reverse Transcription Kit. qPCR was performed using a StepOne Real Time PCR System (Applied Bioscience) and PrimeTime qPCR primer/probe assays, as detailed in the Supplementary Methods (Integrated DNA Technologies). Relative gene expression ($2^{-\Delta\Delta Ct}$) was calculated by normalizing *abcc9* expression to the mean expression observed in WT pools using *Gapdh* as a reference gene and the Pfaffl Method[65].

**Isolation of adult zebrafish ventricular myocytes.** For isolation of cardiomyocytes, ~1-year-old animals were anaesthetized by transfer to ice water before they were decapitated. After removal of the skin and opening of the pericardial sac, the hearts were harvested and both bulbous arteriosus and atrium were removed. Ventricles from three to four fish were pooled together and placed in an epi containing perfusion buffer: (mM) 10 HEPES, 30 Taurine, 5.5 Glucose, 10 BDM in PBS. Ventricular myocytes were obtained by enzymatic dissociation (perfusion buffer supplemented with 12.5 μM $CaCl_2$, 5 mg/ml collagenase II and IV) at 32 °C, and 800 rpm for 40 min. Dissociation was ended by transfer to stopping buffer (perfusion buffer supplemented with 10% (vol/vol) FBS, 12.5 μM $CaCl_2$, 5 mg/ml BSA). Cells were dispersed by gentle trituration using a Pasteur pipette.

**Electrophysiological recordings of isolated myocytes**. Pipettes were made from soda lime glass microhematocrit tubes (Kimble) and had a resistance of 1–2 MΩ when filled with pipette solution. All recordings were made in symmetrical KINT solution ((mM) 140 KCl, 10 HEPES, 0.5 EGTA (pH 7.4 with KOH)). ATP (Sigma) was applied as indicated. Free $Mg^{2+}$ was maintained at 0.5 mM by supplementing ATP-containing solutions with $MgCl_2$ where necessary. Currents were recorded at −50 mV, sampled at 3 kHz, and filtered at 1 kHz using an Axopatch 1D and Digidata 1322 A (Molecular Devices). $K_{ATP}$ current was calculated as the difference between the current in the absence of nucleotides and in the presence of a fully inhibiting concentration of 5 mM ATP. The data were analyzed using Clampfit (Molecular Devices) and Excel (Microsoft), and are presented as individual replicates in scatter plots, alongside mean ± SEM.

**Larval locomotor assay**. The Viewpoint Zebrabox system (Viewpoint Behaviour Technology, Lyon, France) was used to evaluate larval locomotor activity in 5 dpf zebrafish larvae. Larvae were transferred to a 48-well plate with 500 μl of embryo medium, placed into the Zebrabox and subjected to a 30 min acclimatization period to the plate and Zebrabox environment. A total of 24 replicates (one embryo placed individually in each plate well per replicate) were used per genotype. Behavior was monitored at 28.5° using a ZebraBox system (ViewPoint Behaviour Technology) consisting of a soundproof chamber with an infrared camera capable of recording 60 frames per second; analyses were performed using Zebralab locomotion tracking software (ViewPoint Behavior Technology). The integration period was set to 30 s for the 10-min duration of the experiment. A nontransparent background mode with a detection threshold of 20 was used and a minimum velocity of 2 mm s$^{-1}$ was defined as record threshold of inactivity in order to remove system noise. A movement was considered "small" when individuals moved <10 mm/s. The larval locomotor assay consisted of a 5 min baseline recording. The total movement was recorded, then, quantified using ZebraLab software (Viewpoint, Lyon, France) and plotted in "actinteg" units, which is the sum of all pixel changes detected during the experimental period[66]. In addition, from the data recorded, the ZebraLab software calculated several parameters such as total swimming distance (TSD) and total swimming time (TST). The data were compiled and mean activity, TSD or TST for every genotype during each integration period calculated. The data were pooled together from three independent experiments at the same conditions. All experiments were performed comparing SUR2-STOP and its WT siblings. All embryos analyzed originated from group matings of adult zebrafish.

**In vivo high-speed imaging**. Image acquisition was conducted using a Hamamatsu C9300-221 high-speed CCD camera (Hamamatsu Photonics) at 150 fps mounted on a Leica DM IRBE inverted microscope (Leica Microsystems) using Hokawo 2.1 imaging software (Hamamatsu Photonics). Image analysis was subsequently carried out with ImageJ (http://rsbweb.nih.gov/ij/, last accessed November 2017). For analysis of cardiac function, adult zebrafish heterozygous for abcc9 KO were interbred, and genotype identified by sanger sequencing post imaging. High-speed brightfield image sequences of the embryonic zebrafish heart were acquired for zebrafish at age of 5 days post fertilization (dpf). To inhibit pigmentation, 0.003 %(v/v) 1-phenyl-2-thiourea was added to the embryo medium at the Prim-5 stage (about 24 h post fertilization (hpf)). After being anaesthetized in 16 mg/ml tricaine (MilliporeSigma) in E3 medium zebrafish embryos were mounted in dorsal position in small microscopic chambers filled with 0.25% (w/v) agarose (Invitrogen) prepared in the same concentration of anesthetic. Zebrafish hearts were imaged for 10 s (~30 cardiac cycles) at 28 °C ± 0.2 °C.

**Cardiac dimensions and function in SUR2-STOP zebrafish**. The time interval between three heart beats was measured, and the heart rate (bpm) was calculated in triplicates. Sequential still frames from high-speed movies were used to outline the perimeter of the ventricle. Measurement analysis was carried out by "fit-to-ellipse" algorithm, which calculates the center of mass and subsequently the best fitting ellipse. The long axis length (a) and short axis length (b) at diastole and systole were determined and used to calculate ventricular end-systolic (ESV) and diastolic volumes (EDV) in triplicates applying the formula: $V = 4/3\pi a b^2$. The stroke volume was calculated as the difference between three ventricular EDVs and ESVs. Cardiac output was obtained by multiplying the heart rate with stroke volume. The percent shortening fraction (SF) was calculated using the formula: SF = (length at diastole − length at systole)/(length at diastole) × 100. Fractional area change (FAC) was calculated as follows: FAC = (area at diastole − area at systole)/(area at diastole) × 100. Measurements were performed blindly.

**Cardinal vein blood flow velocity**. Zebrafish larvae were mounted in lateral position at 5 dpf and the cardinal vein region behind the cloaca was imaged for 5 s at 28 ± 0.3 °C. Vein blood flow velocity was calculated by assessing frame-by-frame motion of three single erythrocytes per fish determined from high-speed images to assess mean erythrocyte cell velocity (mm/second) using ImageJ. Cardinal blood flow velocity was measured over ten frames at a video frame rate of 150 fps as non-pulsatory venous flow allows frame-by-frame analysis[67].

**Heart extraction and hematoxylin and eosin (H&E) staining**. Adult zebrafish hearts were dissected and fixed in 4% paraformaldehyde (in phosphate buffer with 4% sucrose) at 4 °C overnight and subjected to paraffin embedding and sectioning at 10 μm intervals. Heart sections were stained with H&E following standard procedures. Six fishes were used for each genotype. Image acquisition was conducted using a Leica DM4000 B LED upright automated microscope.

**TUNEL assay**. TUNEL apoptosis staining was performed on 10 μm cryosections and detected using the in situ cell death detection kit from Roche (Mannheim, Germany) according to the manufacturer's instructions. Nuclei are shown by DAPI (4′,6-diamidino-2-phenylindole) staining. Confocal images were acquired using a Leica Sp8 confocal microscope.

**Statistical analysis**. Rubidium flux, patch clamp electrophysiology and echocardiography data were analyzed using two-tailed t test or Mann–Whitney U test as described. For the inverted screen test of mice, data were analyzed using a repeated measures (rm) ANOVA model that contained one between-subjects variable (genotype) and two within-subjects variables (trials and sessions), followed by pairwise comparison by two-tailed t test with Bonferroni correction. Behavioral tests in mice were analyzed using a repeated measures (rm) ANOVA model that contained two between-subjects variable (genotype and sex) and one within-subjects variable (trial block). The data are presented as mean ± SEM.

For zebrafish studies sample size was not predetermined by statistical analysis. In all experiments involving zebrafish embryos, selection was random for scoring. Exact numbers of analyzed embryos are reported at relevant locations in the main text or figure legends. Statistical analysis was carried out with Prism (GraphPad). Distribution of the data sets was tested by D'Agostino–Pearson omnibus normality test. Depending on the outcome of this test, comparison of two conditions with each other was carried out using a Student's two-tailed t test or Mann–Whitney two-tailed U test throughout the manuscript. All values are expressed as mean ± SEM.

**Web resources**. For CHOPCHOP, see http://chopchop.cbu.uib.no/; for Ensembl, see http://www.ensembl.org/; for ExAC Browser, see http://exac.broadinstitute.org/; for gnomAD, see https://gnomad.broadinstitute.org/; for OMIM, see https://www.omim.org/; for UCSC, see https://genome.ucsc.edu/.

**Reporting summary**. Further information on research design is available in the Nature Research Reporting Summary linked to this article.

## Data availability
The authors declare that all data supporting the findings of this study are available within the paper and its Supplementary Information. Source data for Figs. 3, 4, 5, 6, and 7, and Supplementary Figs. 5, 6, 7, and 9 are provided with the paper. The variant described in this study was entered in the Leiden Open Variation Database (LOVD) under ABCC9 (variant #0000579068). The consent does not cover the deposition of the WGS data in a public database, however, this data set is available for academic researchers on request (g.vanhaaften@umcutrecht.nl).

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

## Acknowledgements

First, we thank the patients and families for their participation and consent to publish this work. Help from the following was greatly appreciated: Radiologist Marit Herder (MRI descriptions). Neurologist Synnøve Jensen (neurological examinations), geneticists Ragnhild Glad and Valeria Marton (initial evaluation and diagnostics in the two families). Cardiologists Assami Rösner and Eivind Øygard Fosse (discussion of cardiac features), ENT specialist Dagny Hemmingsen (hearing evaluation), the National Neuromuscular Center, Tromsø, Norway. We acknowledge the support from the E-Rare Joint Transnational Cantú Treat program (I-2101-B26), and NIH grant HL140024 (to CGN). Partial support for the behavioral work was provided by the Intellectual and Developmental Disabilities Research Center at Washington University (NIH/NICHD U54 HD087011; DFW). C.Mc. is supported by American Heart Association Fellowship (19POST34380407).

## Author contributions

M.F.S. identified and matched the study subjects. M.F.S., K.A.A., K.I.M., and A.R.D. performed clinical patient examinations. M.F.S., C.Mc., H.I.R., D.F.W., M.V.G., C.G.N., and G.v.H. designed the study and formulated the research question. GAMH and MM carried out the formal analysis. T.H., M.S.R., S.S., J.B., and F.T. created the animal models. H.H., M.V.G., C.Mc., H.I.R., M.S.R., C.H.E., A.K., Y.H., S.S.S., J.G., D.F.W., and S.M.K. performed the experiments. M.F.S., C.Mc., H.I.R., G.A.M.H., and A.R.D. wrote the initial paper draft, and M.F.S., C.Mc., H.I.R., M.V.G., C.G.N., and G.v.H. reviewed and revised the paper. G.A.M.H. critically reviewed the paper. G.v.H. and C.G.N. provided funding acquisition.

## Competing interests

The authors declare no competing interests.
