## [Peer Review File · Nature Communications]

Reviewers' comments:

Reviewer #1 (Remarks to the Author):

In this study, Smeland and co-workers link a mutation in the ABCC9 gene with a syndrome encompassing amongst others intellectual disability and myopathy. The genetic data is convincing. The reviewer does have concerns about the fact that the genetic models (mouse and fish models) do not faithfully copy the genetic defect of the patients. In essence, while abnormal SUR2A protein (lacking 52 amino acids due to the inframe exclusion of exon 8) is presumed to be present in the patients as suggested by the mRNA studies in fibroblasts, the two models that are studied essentially lack the SUR2A protein completely. On the other hand, the knock-out mice and zebrafish do recapitulate important clinical features observed in the patients, supporting a possible loss of function mechanism.

It is unclear why the authors chose for two animal models rather than conducting more in-depth phenotypic and mechanistic studies in one model. For example the histological characterization of the heart in the fish is rather superficial (how is cardiomyocyte size? How are the myofibers? Is there cell death? Is there fibrosis? Similarly, the electrophysiological phenotype in skeletal muscle and cardiac muscle is unexplored... Also, no histology of skeletal muscle is presented. Cerebral white matter hyperintensity seems to be a consistent feature of the disorder. Did the authors consider conducting histological studies in the mouse brains?

The authors should make an effort to determine whether phenotypic features observed in the fish (e.g. the hypotelorism) was also observed in the mice. Similarly, whether the thickened myocardium observed in the fish is also observed in the mice.

Minor comments:

What was the genetic background of the founder SUR2-STOP Mouse? Please mention this in the Methods.

Line 441: Please specify the purpose of the Single Nucleotide Polymorphism (SNP) array.

Line 442: 'sequencing of multiple neuromuscular disease genes, screening for mitochondrial DNA sequence variants/deletions and screening for inborn errors of metabolism.' How does this relate to the TruSightOne gene panel analysis?

Line 475: what is meant by 'mainly healthy'?

Line 476: Something seems missing here: 'Weight and length were below the 2.5% in childhood.'

Line 530: '530 Individual II-3 was a female fetus, conceived between patients 1-2 and 1-3.' There is here a mistake in the numbering/nomenclature.

Line 562: 562 Patient 2-1 > 562 Patient 2-2. Please check all the numberings. There are other mistakes besides the ones mentioned.

Line 609: 'but not identical variants,' Do you mean 'not necessarily identical variants'?

Line 622: Please provide GNOMAD frequency instead ExAC. The c.1320+1G>A variant is present 25

times in Gnomad (no homozygotes). Please mention this and discuss.

Reviewer #2 (Remarks to the Author):

The manuscript describes six patients from two families, phenotypically characterized by mild intellectual disability, similar facial features, myopathy, cerebral white matter hyperintensities and cardiac systolic dysfunction, and genotypically identified by homozygous splice variant ABCC9 variant that results in aberrant protein structure and function. The authors propose to name this novel ABCC9 channelopathy ABCC9-related Intellectual disability Myopathy Syndrome (or AIMS). Kir6.2/SUR2 channels carrying the deleted sequence fail to efflux radioactive rubidium from transfected cells. Mice and zebrafish carrying various stop codon insertions and early terminations are used as model systems to illustrate that ABCC9-null animals have cardiac defects, but do not recapitulate most of the phenotypes observed in AIMS.

A major strength of the study is the description of a syndrome that affects two Finish families. If this syndrome is indeed monogenic and related solely to the homozygous ABCC9 c.1320+1G>A splice donor variant, then this syndrome might be more prevalent than suspected. The reason is that, according to GnomAd, the heterozygous 12-22063090-C-T variant (or rs1362571466 in dbSNP) is not all that rare, and is present predominantly in the Finish population (18/24,850 alleles, with an allele frequency of 0.0007%), less common in non-Finnish Europeans, and absent in Asian or African populations (<https://gnomad.broadinstitute.org/variant/12-22063090-C-T>). The manuscript will benefit from the inclusion of these data.

The WGS data analysis has identified 67 shared variants in two of the index patients studied. Gene interactions with the homozygous ABCC9 c.1320+1G>A variant cannot be excluded as participating in the syndrome, especially given that almost 1 in every 1000 Finish individuals are heterozygous for the ABCC9 variant (according to GnomAd at least). At the minimum, this possibility should be mentioned and the shared variants, along with their allele frequencies, should be provided in the Supplemental text (it appears that only some are displayed).

Several other ABCC9 splice variants have been described and it may be useful to mention these in line 65.

Ensembl indicates exon 8 to code for SUR2 amino acids A389-Q440 (A M I Y N K I L R L S T S N L S M G E M T L G Q I N N L V A I E T N Q L M W F L F L C P N L W A M P V Q) - http://useast.ensembl.org/Homo_sapiens/Transcript/Exons?db=core;g=ENSG00000069431;r=12:21797401-21942529;t=ENST00000261200. Please verify the numbering of Ala386 to Gln437 used in line 204.

Please indicate the position of the SUR2 flag tag (N-terminal?) (line 217)

Two SUR2-STOP animal models were used with genotypes that differ from human AIMS. This is regrettable, but understandable given the cost and time it would take to generate AIMS-relevant animal models (which likely will also lack functional SUR2 proteins). It should be made more clear, though, that these animals were not used by design, but rather because they were already at hand and useful to study the SUR2-null cardiovascular and behavioral phenotypes.

It is curious that different methods are used to assess KATP channel function in Figures 3 and 4. The more relevant patch clamp methods are absent for SUR2-delta8. Given the absence of Rb flux for the homozygous SUR2-delta8, one would not expect much by the way of channel function. Patch clamp data would be helpful, though, to determine whether the heterozygous SUR2/SUR2-delta8 channel is normal in every respect (nucleotide sensitivity and pharmacology, for example). One is also left wondering, in Fig 3, whether the absence of SUR2-delta8 flux is due to the absence of surface channels (the most likely explanation) or whether non-functional channels reached the membrane. It is also curious that 50% of wild-type SUR2 fully rescues Rb flux, which not only is relevant to the AIMS parents, but also for the concept that SUR2 is the rate-limiting subunit that determines KATP channel expression (PMID: 19084477). If not addressed experimentally, these issues should be discussed.

Were there any cardiovascular phenotypes in the heterozygote SUR2-STOP animals? Also, since SUR2 has a major contribution to smooth muscle function, did the homozygote SUR2-STOP animals have obvious smooth muscle phenotypes? Please mention during revision.

Reviewer #3 (Remarks to the Author):

Smeland and colleagues report a novel syndrome associated with homozygous loss of function of the ABCC9 gene that encodes the SUR2 regulatory subunit of I-KATP channels in 2 Northern European families. The families apparently are not closely related, but in each case, the parents both harbor a ultra-rare splice variant predicted to cause a premature stop codon. The authors provide detailed experiments to confirm the loss of function nature of this disorder in a heterologous expression system, as well as zebrafish and murine models. For the most part, the animal models recapitulate features of the human phenotype. I have no major concerns regarding this study and in fact the authors provide compelling and detailed analyses of the clinical and experimental features. That being said, I reviewed the videos of the 5-day WT and homozygous null ABCC9 zebrafish cardiac imaging studies and I am not convinced that there is much of a difference in cardiac performance by visual inspection. In light of this, I recommend the authors temper their description of a true cardiac phenotypes at 5 dpf. The evidence from cardiac pathology of older zebrafish is very clear; similar to the adult onset cardiac dysfunction.

Mentioned edits/additions are included in the manuscript in yellow highlight.

Reviewers' comments:

Reviewer #1 (Remarks to the Author):

1. In this study, Smeland and co-workers link a mutation in the *ABCC9* gene with a syndrome encompassing amongst others intellectual disability and myopathy.

The genetic data is convincing. The reviewer does have concerns about the fact that the genetic models (mouse and fish models) do not faithfully copy the genetic defect of the patients. In essence, while abnormal SUR2A protein (lacking 52 amino acids due to the inframe exclusion of exon 8) is presumed to be present in the patients as suggested by the mRNA studies in fibroblasts, the two models that are studied essentially lack the SUR2A protein completely. On the other hand, the knock-out mice and zebrafish do recapitulate important clinical features observed in the patients, supporting a possible loss of function mechanism.

We appreciate the detailed and constructive comments from the reviewer, and accept and acknowledge the fact that the animal models are not a genetic recapitulation of the human context. As suggested by reviewer 2 – the cost and time required to generate novel models is limiting. The studies of recombinant SUR2Δ8 containing channels provide strong evidence that the exon deletion results in complete loss of KATP channel function, and so the functional consequences of the mutations in AIMS patients is mirrored by the functional consequences in the animal models. We appreciate the recognition and agree that the animal models recapitulate important clinical features, suggesting that the conserved functional consequences in humans, mice and fish underlie the clinical pathology.

Whilst we appreciate the concern the reviewer states, we are of the opinion the insights provided by the SUR2-STOP animals are useful and timely at this stage.

Also, as described in comment 4, we added further evidence for a complete loss of KATP channel function in the applied zebrafish model by recording channel activity in cardiomyocytes derived from WT and SUR2-STOP fish. These results confirm abolishment of functional KATP expression in SUR2-STOP hearts as shown in recombinant channels and SUR2-STOP mice.

2. It is unclear why the authors chose for two animal models rather than conducting more in-depth phenotypic and mechanistic studies in one model.

In addition to the increased confidence that recapitulating key clinical features in 2 distinct animal models provides for determining that SUR2 LoF underlies AIMS, the phenotypic characterization we describe helps to establish the SUR2-STOP mice and zebrafish as useful tools for future study of AIMS. For example, the identification of conserved cardiovascular and motility/fatigability phenotypes in humans, mice and fish allow for the study of corrective effects of potential future pharmacotherapies. Demonstrating recapitulation in both models allows for capitalization on the relative advantages of mouse (mammalian physiology) and zebrafish (amenability to high throughput screening) models. To address this, we have added the following statement (pg 43 line 1044):

“Here, we demonstrate that key cardiovascular and motility/fatigability features of AIMS are conserved in both SUR2-STOP mice and zebrafish. These data establish the animal models as key tools for the future study of the pathophysiological mechanisms underlying AIMS features and for the investigation of potential pharmacotherapies.”

3. For example the histological characterization of the heart in the fish is rather superficial (how is cardiomyocyte size? How are the myofibers? Is there cell death? Is there fibrosis?)

We agree with the reviewer and have performed histological characterization of the fish myocardium as suggested. We have now isolated six hearts of both WT and SUR2-STOP fish, embedded in tissue freezing medium and performed sectioning. Firstly, cryosections were used to perform Acid Fuchsin Orange G (AFOG) staining which labels the myocardium orange, collagen blue and fibrin red. There was no fibrosis visible in SUR2-STOP hearts with both ventricle and atrium looking unaltered from WT hearts (Figure S11).

Next, we subjected cryosections to immunohistochemistry. To address possible cell apoptosis in SUR2-STOP hearts, we applied TUNEL (TdT-mediated nick end labeling). While we could only very rarely detect apoptotic cells in WT hearts, a sizable number of cells was TUNEL-positive in both cardiac chambers of SUR2-STOP fish (Figure 7H).

Lastly, we confirmed an unaltered myofiber structure by performing immunohistochemistry staining tropomyosin. We assessed four different areas of both WT and SUR2-STOP hearts: center of ventricle, apex, ventricular border and atrium. However, no differences could be observed.

We have added reference to this new data (pg 36 line 849):

“In order to further characterize heart histology, we stained cryosections of WT and SUR2-STOP hearts with Acid Fuchsin Orange G (AFOG), which labels myocardium orange, collagen blue and fibrin red indicating no visible fibrosis in both ventricle and atrium of SUR2-STOP fish (Figure S11). To address possible heart apoptosis TUNEL (TdT-mediated nick end labeling) was performed. While we could only very rarely detect apoptotic cells in WT hearts, a sizable number of cells was TUNEL-positive in both cardiac chambers of SUR2-STOP fish (Figure 7H). Lastly, we confirmed that myofiber structure in SUR2-STOP hearts is unaltered from WT performing immunohistochemistry staining tropomyosin (Figure S12).”

Unfortunately, it was not possible for us to analyze cardiomyocyte size in adult WT and SUR2-STOP hearts since we are currently lacking an adequate technique to investigate this question sufficiently. We considered two different approaches: 1) immunofluorescence on cryosections staining the membrane of cardiomyocytes to visualize their size and 2) staining cardiomyocyte nuclei on cryosections to quantify the distance between nuclei to get an impression on cell size. However, we had a number of major concerns regarding both approaches. For instance, from experience we know that tissue frozen as cryosections can slightly expand when thawing which would have a direct effect on the original cardiomyocyte size.

Notably, both techniques have been tried in the past to assess cardiomyocyte size in a different zebrafish model in our lab and proved to not yield conclusive and satisfactory results.

4. Similarly, the electrophysiological phenotype in skeletal muscle and cardiac muscle is unexplored...

We have now isolated cardiomyocytes from the zebrafish heart and recorded KATP channel activity from WT and SUR2-STOP fish. As we now show in Fig. 6D and E, functional KATP expression is abolished in SUR2-STOP cardiomyocytes. We have added reference to this new data (pg 34 line 805):

“Furthermore, inside-out patch clamp recordings from acutely isolated ventricular myocytes from the SUR2-STOP zebrafish revealed a complete absence of functional KATP channels, in comparison to WT controls (Figure 6D).”

Unfortunately, at this stage we have not established electrophysiological methods for recording from skeletal muscle, though this is an interesting direction for future study.

5. Also, no histology of skeletal muscle is presented.

We acknowledge the lack of skeletal muscle histology from animal models. Notably, a muscle biopsy was sampled from one patient (Patient 1-2) which revealed only minor histological changes (unspecific mitochondrial aggregation and muscle fiber caliber variation). Interestingly, previous studies of KATP knockout mice have reported little evidence of structural or histological abnormalities in non-exercised animals which is consistent with the apparently minor structural abnormalities in the patient sample and may be suggestive of only relatively minor structural abnormalities

resulting from KATP knockout (Thabet et al., *Physiological Genomics*, 2005), and so this was not explored further. We have included reference to this in the manuscript (pg 38 line 918):

“Notably, only relatively minor histological abnormalities were observed in a single skeletal muscle biopsy from Patient 1-2. Interestingly, this is consistent with previous reports which showed that SUR2 knockout mice only exhibit significant histological abnormalities when subjected to significant and chronic exercise (Thabet et al., *Physiological Genomics*, 2005)”

6. Cerebral white matter hyperintensity seems to be a consistent feature of the disorder. Did the authors consider conducting histological studies in the mouse brains?

Thank you for the astute and helpful observation. We agree that the white matter hyperintensity is a prominent feature of the disorder. It would be fascinating to study this further in the animal models, unfortunately however, observing white matter lesions in mice is notoriously difficult as mice have very little white matter (there is no animal model for white matter disease). Nonetheless, future studies of mouse brain histology from SUR2-STOP mice would indeed be interesting to explore this further.

7. The authors should make an effort to determine whether phenotypic features observed in the fish (e.g. the hypotelorism) was also observed in the mice.

Prompted by the reviewer’s comments we have measured inter-eye distance and see no significant difference between WT and SUR2-STOP mice, we have now included reference to this (pg 34 ln 812):

“In contrast, when the inter-eye distance was measured in mice, no significant difference was observed between WT and SUR2-STOP mice.”

8. Similarly, whether the thickened myocardium observed in the fish is also observed in the mice.

Thanks you for this suggestion. We have now performed analysis of heart size in SUR2-STOP mice. Interestingly, in echocardiographic measurements we observe a small, but statistically significant increase in left-ventricular internal dimension in diastole. This mirrors the mild LV dilatation observed our eldest patient (Patient 2-1), and may be reflective of the dilated cardiomyopathy previously associated with SUR2 mutations (Bienegraeber et al., *Nature Genetics*, 2004). We thank the reviewer for this helpful suggestion and have added these data to the paper (pg 33 lin 769):

“Furthermore a small, but statistically significant increase in left-ventricular internal dimension in diastole (normalized to body length) was observed in SUR2-STOP mice, which mirrors the mild LV dilatation observed our eldest patient (Patient 2-1), and may be reflective of the dilated cardiomyopathy previously associated with SUR2 mutations (Bienegraeber et al., *Nature Genetics*, 2004).”

Echo analysis showed an increase in left ventricular mass (indexed to body weight) in SUR2-STOP mice, which is consistent with previous studies of heart weight normalized to body weight in SUR2 KO mice (Stoller et al., *JMCC* 2007; Gau et al., *Clinical Proteomics*, 2014). However, as also reported previously, we observed decreased body weight in SUR2-STOP mice, which means the apparent increase in left ventricle mass we observed (and normalized heart weight previously reported) may in fact be a distortion introduced by normalization by body weight. When we normalize isolated heart weight to mouse body length, we observed a trend towards an increase in mass, but this was not statistically significant (as shown in Figure 4).

9. What was the genetic background of the founder SUR2-STOP Mouse? Please mention this in the Methods.

B6CBA F1/J. We have added reference to this (pg 12 ln 287).

10. Line 441: Please specify the purpose of the Single Nucleotide Polymorphism (SNP) array.

Earlier clinical investigations identifying mild developmental delay in affected family members prompted genetic testing. As a first line investigation standard G-banding and FMR1 CGG repeat analysis was performed. Additionally, high resolution Single Nucleotide Polymorphism (SNP) array was done to investigate for smaller deletions/duplications. This was specified in the manuscript as follows (pg. 22 ln. 581):

“Genetic investigations had earlier been performed with normal results in several of the patients, including G-banding, high-resolution Single Nucleotide Polymorphism (SNP) array to look for genomic deletions and duplications, FMRI CGG repeat analysis, DMPK PCR, sequencing of multiple neuromuscular disease genes, screening for mitochondrial DNA sequence variants/deletions and screening for inborn errors of metabolism.”

11. Line 442: ‘sequencing of multiple neuromuscular disease genes, screening for mitochondrial DNA sequence variants/deletions and screening for inborn errors of metabolism.’ How does this relate to the TruSightOne gene panel analysis?

Due to the muscular weakness present in the patients, multiple neuromuscular disease genes were analyzed both by sequencing or RP-PCR. In search for the molecular cause of the disorder in these families, mitochondrial DNA was studied for sequence variants and deletions and screening for inborn metabolic disorders was also performed. The collection of above-mentioned investigations gave normal results. These normal results led us to investigate patients for mutations in Mendelian genes included in TruSightOne gene panel.

12. Line 475: what is meant by ‘mainly healthy’?

We changed this into “healthy”.

13. Line 476: Something seems missing here: ‘Weight and length were below the 2.5% in childhood.’

Percentile, now revised.

14. Line 530: ‘530 Individual II-3 was a female fetus, conceived between patients 1-2 and 1-3.’ There is here a mistake in the numbering/nomenclature.

We kindly refer to the pedigree in Figure 1A in order to explain the numbering and nomenclature. We used two different ways of numbering patients (for instance patient 1-2 and 1-2) and family members (for instance individual I.1 and II.3). Hence, the numbering is correct.

In order to avoid confusion, we changed the nomenclature of the patients in Figure 2B to 1-2, 1-3 and 1-4.

15. Line 562: 562 Patient 2-1 > 562 Patient 2-2. Please check all the numberings. There are other mistakes besides the ones mentioned.

Please see our response to the question above.

16. Line 609: 'but not identical variants,' Do you mean 'not necessarily identical variants'?

We are referring to pathogenic variants that can be found in the same gene and lead to the same clinical phenotype. To make this clearer, we adapted the manuscript accordingly (pg 29 ln 672):

"The focus of the analysis was the identification of shared variants or different variants in shared genes with possible damaging, but not identical variants leading to the same clinical phenotype, amongst the two cases."

17. Line 622: Please provide GNOMAD frequency instead ExAC. The c.1320+1G>A variant is present 25 times in Gnomad (no homozygotes). Please mention this and discuss.

Thank you for this suggestion, please see our response to the similar comment from Reviewer 2.

Reviewer #2:

The manuscript describes six patients from two families, phenotypically characterized by mild intellectual disability, similar facial features, myopathy, cerebral white matter hyperintensities and cardiac systolic dysfunction, and genotypically identified by homozygous splice variant ABCC9 variant that results in aberrant protein structure and function. The authors propose to name this novel ABCC9 channelopathy ABCC9-related Intellectual disability Myopathy Syndrome (or AIMS). Kir6.2/SUR2 channels carrying the deleted sequence fail to efflux radioactive rubidium from transfected cells. Mice and zebrafish carrying various stop codon insertions and early terminations are used as model systems to illustrate that ABCC9-null animals have cardiac defects, but do not recapitulate most of the phenotypes observed in AIMS.

1. A major strength of the study is the description of a syndrome that affects two Finish families. If this syndrome is indeed monogenic and related solely to the homozygous ABCC9 c.1320+1G>A splice donor variant, then this syndrome might be more prevalent than suspected. The reason is that, according to GnomAd, the heterozygous 12-22063090-C-T variant (or rs1362571466 in dbSNP) is not all that rare, and is present predominantly in the Finish population (18/24,850 alleles, with an allele frequency of 0.0007%), less common in non-Finish Europeans, and absent in Asian or African populations (<https://gnomad.broadinstitute.org/variant/12-22063090-C-T>). The manuscript will benefit from the inclusion of these data.

Thank you for the positive and insightful comments. We appreciate the reviewer's sharp observation of the not-all-that-rare variant frequency, and agree that this may indeed suggest that this disorder is not as rare as it first appears, particularly in Finnish populations. We have now included reference to this in the manuscript (pg 30 ln 692):

"Notably, according to gnomAD, the variant is reported at relatively high frequency in heterozygous state in the Finnish population with an allele frequency of 0.0007 (18/24.850). It is less common in non-Finish Europeans (5/128.232, allele frequency 0.00004) and absent in Asian or African populations (May 2019). Considering the probable Finnish ancestry of all patients, the syndrome might be more prevalent in the Finnish population than suspected. The homozygous state is absent in gnomAD."

2. The WGS data analysis has identified 67 shared variants in two of the index patients studied. Gene interactions with the homozygous ABCC9 c.1320+1G>A variant cannot be excluded as participating in the syndrome, especially given that almost 1 in every 1000 Finish individuals are heterozygous for the ABCC9 variant (according to GnomAd at least). At the minimum, this possibility should be mentioned and the shared variants, along with their allele frequencies, should be provided in the Supplemental text (it appears that only some are displayed).

Thank you for this comment. Indeed, we agree that gene interactions with the homozygous variant in ABCC9 cannot be excluded. Additionally, we provide a table indicating all 67 shared variants found in the WGS analysis and their allele frequencies according to 1000 genomes and ExAC. Please note, that we performed our analysis with the Alissa cohort tool where gnomAD is not annotated. We included the following to the manuscript (pg 29 In 680):

“Notably, gene interactions with the homozygous ABCC9 c.1320+1G>A variant cannot be excluded as participating in the syndrome. A list of all shared variants including allele frequencies is provided in Table S2.”

3. Several other ABCC9 splice variants have been described and it may be useful to mention these in line 65.

We have added reference to these additional splice variants.

“– whilst multiple other splice variants have also been reported (Ye et al., 2009; Shi et al., 2005; Chutkow et al., 1999).

4. Ensembl indicates exon 8 to code for SUR2 amino acids A389-Q440 (A M I Y N K I L R L S T S N L S M G E M T L G Q I N N L V A I E T N Q L M W F L F L C P N L W A M P V Q) -

http://useast.ensembl.org/Homo_sapiens/Transcript/Exons?db=core;q=ENSG00000069431;r=12:21797401-21942529;t=ENST00000261200. Please verify the numbering of Ala386 to Gln437 used in line 204.

You are indeed correct that exon 8 encodes A389-Q440 in the human sequence. The recombinant constructs used for functional studies were rat SUR2A and SUR2B, which vary very slightly from the human sequence (3 amino acids are absent in the rat sequence, accounting for the numbering difference), but the overall sequence identity is very high (97%). The rat SUR2A and SUR2B clones have been used extensively in previous studies of recombinant KATP channels.

5. Please indicate the position of the SUR2 flag tag (N-terminal?) (line 217)

Yes, this was an N-terminal Flag tag, now included in methods: “(N-terminal Flag tag with amino acid sequence MDYKDDDDKGAP was inserted prior to the SUR2A start codon)”

Two SUR2-STOP animal models were used with genotypes that differ from human AIMS. This is regrettable, but understandable given the cost and time it would take to generate AIMS-relevant animal models (which likely will also lack functional SUR2 proteins). It should be made more clear, though, that these animals were not used by design, but rather because they were already at hand and useful to study the SUR2-null cardiovascular and behavioral phenotypes.

We appreciate this helpful comment, and have included the following addition to the discussion (pg 37 line 876):

“The animal models used in this study do not recapitulate the genetic defect identified in the AIMS patients, but were chosen as the functional effects of the frame-shift mutations introduced into SUR2-STOP mice and fish mirror the functional effect of the SUR2Δ8 mutation. Future studies of CRISPR/Cas9 genome edited animal models in which human-disease associated AIMS mutations are introduced may provide further insights into the severity and variety of phenotypes arising from specific mutations.”

6. It is curious that different methods are used to assess KATP channel function in Figures 3 and 4. The more relevant patch clamp methods are absent for SUR2-delta8. Given the absence of Rb flux for the homozygous SUR2-delta8, one would not expect much by the way of channel function. Patch clamp data would be helpful, though, to determine whether the heterozygous SUR2/SUR2-delta8 channel is normal in every respect (nucleotide sensitivity and pharmacology, for example). One is also left wondering, in Fig 3, whether the absence of SUR2-delta8 flux is due to the absence of surface channels (the most likely explanation) or whether non-functional channels reached the membrane. It is also curious that 50% of wild-type SUR2 fully rescues Rb flux, which not only is relevant to the AIMS parents, but also

for the concept that SUR2 is the rate-limiting subunit that determines KATP channel expression (PMID: 19084477). If not addressed experimentally, these issues should be discussed.

Thank you for these suggestions, which we have commented on as below. We have now performed patch clamp analysis of recombinant KATP channels, and included this novel data (Fig. 3E-G). These experiments show that, as expected, no functional channels are observed in the “homozygous” context. Prompted by the reviewer’s suggestion, we have also examined the effect of the co-expression of WT and SUR2-delta8 subunits on nucleotide sensitivity and pharmacological activation by the KATP channel opener pinacidil, and show that there is no significant effect of the mutation in this context. We have included reference to these data (pg 31 line 726):

“Consistent with flux experiments, a complete absence of functional KATP channels was observed in inside-out patch clamp recordings from cells co-transfected with Kir6.2 alongside SUR2AΔ8, in contrast to robust expression in cells transfected with SUR2A-WT or a 1:1 mix of SUR2A-WT and SUR2AΔ8 (Figure 3E-F). Therefore, homomeric deletion of exon 8 results in a significant decrease in protein expression and complete loss of KATP channel function. Co-expression of SUR2A-WT and SUR2AΔ8 did not affect channel regulation by ATP or pharmacological activation by pinacidil, suggesting that in the heterozygous context, the c.1320+1G>A mutation is without significant effect (Figure 3G and S5).”

At this stage we do not know whether the decreased functional expression is due to loss of surface membrane expression, or due to the non-functional channels reaching the membrane which is an interesting point and can be followed up in more detailed molecular studies of this AIMS mutation. We have added a comment to this effect (pg 31 line 731):

“The decrease in functional expression of SUR2Δ8 containing channels may arise due to either the absence of surface-expressed channels or the surface-expression of non-functional channel complexes, which requires more detailed molecular study for elucidation.”

Unfortunately, the drastic over-expression of channels in our heterologous studies make it difficult to completely faithfully model rate-limiting expression. The functional expression of KATP complexes may not linearly correlate with the cDNA amount transfected (and can vary significantly day-to-day with different transient transfections). What is clear from our studies, is that co-expression of SUR2Δ8 with WT cDNA does not have a dominant-negative effect. This helps to explain why heterozygous carriers do not exhibit AIMS pathologies. The reviewer identifies an interestingly idea that functional expression (if indeed SUR2-limited) may be decreased in het carriers, but this would require novel models for study (either new animal models with the Δ8 splice site mutation or perhaps patient/carrier derived hiPSC-cardiomyocyte models) – this would indeed be an interesting future direction.

7. Were there any cardiovascular phenotypes in the heterozygote SUR2-STOP animals?

We performed high-speed video imaging of the hearts of 5dpf heterozygous SUR2-STOP fish as indicated in Supplementary Figure x. We were not able to observe any cardiac abnormalities with ventricular contractility and cardiac output behaving similar to WT animals. Hence, we did not see a reason to proceed with histological analysis of adult heterozygous fish as done with homozygous SUR2-STOP fish. We have added a comment to these findings (pg 35 line 843):

“No cardiac abnormalities were observed in 5dpf larvae heterozygous for the SUR2-STOP mutation which behaved like WT controls (Figure S9).”

8. Also, since SUR2 has a major contribution to smooth muscle function, did the homozygote SUR2-STOP animals have obvious smooth muscle phenotypes? Please mention during revision.

Thank you for this suggestion. We have now performed patch clamp studies of acutely isolated aortic smooth muscle

cells and demonstrate a complete abolition of functional KATP currents in SUR2-STOP mice. Further, we have recorded blood pressure in SUR2-STOP mice, and show that KATP LoF results in an ~20mmHg increase in mean arterial pressure, consistent with previous reports of Kir6.1 and SUR2 KO mice (Miki, Nature Medicine, 2002; Chutkow JCI, 2002). We have included these data (Supplemental Fig. 6) and comment (pg 33 line 773).

“In addition, blood pressure was significantly increased in SUR2-STOP mice (Figure S6C-D), consistent with the known role of SUR2-containing KATP channels in vascular smooth muscle cells in the regulation of vascular tone (Chutkow JCI, 2002).”

Furthermore, we performed high-speed video imaging of the cardinal vein in 5dpf wildtype and SUR2-STOP zebrafish larvae to assess vein blood flow velocity. Here, SUR2-STOP larvae revealed an increased red blood cell velocity indicating high blood pressure (Figure 7F). We have included the following comment to the manuscript (pg 35 line 839):

“Additionally, we applied high-speed video imaging of the cardinal vein to assess blood flow velocity of wildtype and SUR2-STOP larvae. An increased velocity of red blood cells in SUR2-STOP fish (Figure 7F) can be associated with high blood pressure found in SUR2-STOP mice.”

Reviewer #3:

Smeland and colleagues report a novel syndrome associated with homozygous loss of function of the ABCC9 gene that encodes the SUR2 regulatory subunit of I-KATP channels in 2 Northern European families. The families apparently are not closely related, but in each case, the parents both harbor a ultra-rare splice variant predicted to cause a premature stop codon. The authors provide detailed experiments to confirm the loss of function nature of this disorder in a heterologous expression system, as well as zebrafish and murine models. For the most part, the animal models recapitulate features of the human phenotype. I have no major concerns regarding this study and in fact the authors provide compelling and detailed analyses of the clinical and experimental features.

1. That being said, I reviewed the videos of the 5-day WT and homozygous null ABCC9 zebrafish cardiac imaging studies and I am not convinced that there is much of a difference in cardiac performance by visual inspection. In light of this, I recommend the authors temper their description of a true cardiac phenotypes at 5 dpf. The evidence from cardiac pathology of older zebrafish is very clear; similar to the adult onset cardiac dysfunction.

We agree with the reviewer that the cardiac phenotype is not easily observed when looking at the supplied movies without further attached explanation. Hence, we added Figure S8 in order to illustrate the analysis of such high-speed imaging movies to examine heart function, especially ventricular contractility which is the primary cardiac phenotype observed in AIMS patients and SUR2-STOP mice. Movies were stopped exactly at diastole, when the ventricular area is at its largest, and at systole, when it is at its smallest size. These still frames were used to outline the perimeter of the ventricle. Measurement analysis was carried out by “fit- to-ellipse” algorithm, which calculates the center of mass and subsequently the best fitting ellipse. The long axis length and short axis length at diastole and systole were determined and used to calculate ventricular end-systolic (ESV) and diastolic volumes (EDV), stroke volume, cardiac output, fractional shortening and ejection fraction. This method is an accepted standard technique to examine heart function in zebrafish larvae and has been applied in various publications (Shin et al., Physiol Genomics, 2010; Yagoob et al., Comp Biochem Physiol A Mol Integr Physiol., 2010; Yalcin et al, Dev Dyn, 2017). Figure S8 shows exemplary still frames from a movie of a WT and SUR2-STOP heart at diastole and systole. Ventricular areas are highlighted by the best fitting ellipse, with the ventricular diameter D_d and D_s indicated by a black arrow. When comparing the difference between D_d and D_s in WT and SUR2-STOP fish, you can see an obvious decrease in contractile ability in mutant fish. As a result, the stroke volume and consequently the cardiac output are also decreased. Hence, this shows that we are dealing with a true cardiac phenotype at 5dpf.

Additionally, all measurements were done in triplicates (3x diastole and 3x systole from one movie) and analysis was done blindly without knowing the genotype in advance, which strengthens the observed features.

REVIEWERS' COMMENTS:

Reviewer #1 (Remarks to the Author):

No further comments.

Reviewer #2 (Remarks to the Author):

The authors have fully addressed each of my concerns. I have no further comments.

Reviewer #3 (Remarks to the Author):

The authors have addressed my comments/concerns.

REVIEWERS' COMMENTS:

Reviewer #1 (Remarks to the Author):

No further comments.

Reviewer #2 (Remarks to the Author):

The authors have fully addressed each of my concerns. I have no further comments.

Reviewer #3 (Remarks to the Author):

The authors have addressed my comments/concerns.